# GeneLab Database Analyses Suggest Long-Term Impact of Space Radiation on the Cardiovascular System by the Activation of *FYN* Through Reactive Oxygen Species

**DOI:** 10.3390/ijms20030661

**Published:** 2019-02-03

**Authors:** Afshin Beheshti, J. Tyson McDonald, Jack Miller, Peter Grabham, Sylvain V. Costes

**Affiliations:** 1WYLE Labs, NASA Ames Research Center, Moffett Field CA 94035, USA; 2Department of Physics, Hampton University, Hampton, VA 23668 USA; john.mcdonald@hamptonu.edu; 3Lawrence Berkeley National Laboratory, Berkeley, CA 94720, USA; j_miller@lbl.gov; 4Center for Radiological Research, Columbia University, New York, NY 10032, USA; pwg2@cumc.columbia.edu; 5NASA Ames Research Center, Space Biosciences Division, Moffett Field, CA 94035, USA

**Keywords:** space radiation, GeneLab, cardiovascular, HUVECs, *FYN*, ROS, reactive oxygen species, cardiomyocytes, protons, iron, HZE irradiation

## Abstract

Space radiation has recently been considered a risk factor for astronauts’ cardiac health. As an example, for the case of how to query and identify datasets within NASA’s GeneLab database and demonstrate the database utility, we used an unbiased systems biology method for identifying key genes/drivers for the contribution of space radiation on the cardiovascular system. This knowledge can contribute to designing appropriate experiments targeting these specific pathways. Microarray data from cardiomyocytes of male C57BL/6 mice followed-up for 28 days after exposure to 900 mGy of 1 GeV proton or 150 mGy of 1 GeV/n ^56^Fe were compared to human endothelial cells (HUVECs) cultured for 7 days on the International Space Station (ISS). We observed common molecular pathways between simulated space radiation and HUVECs flown on the ISS. The analysis suggests *FYN* is the central driver/hub for the cardiovascular response to space radiation: the known oxidative stress induced immediately following radiation would only be transient and would upregulate *FYN*, which in turn would reduce reactive oxygen species (ROS) levels, protecting the cardiovascular system. The transcriptomic signature of exposure to protons was also much closer to the spaceflight signature than ^56^Fe’s signature. To our knowledge, this is the first time GeneLab datasets were utilized to provide potential biological indications that the majority of ions on the ISS are protons, clearly illustrating the power of omics analysis. More generally, this work also demonstrates how to combine animal radiation studies done on the ground and spaceflight studies to evaluate human risk in space.

## 1. Introduction

Long-duration space travel has been associated with a number of human health risk factors, such as space radiation, microgravity, isolation, and hypoxia [1,2,3,4]. It is now thought that the two main factors driving the majority of the health risks are space radiation and microgravity, with space radiation being the main component for health risk for space missions beyond low Earth orbit (LEO) [5,6,7,8,9,10]. The different organs and related health issues potentially induced by these two aspects of the space environment are continuously studied and updated.

Space radiation observed on the International Space Station (ISS) is quite different from ionizing radiation found on Earth and is composed primarily of protons (both high and low energy) and HZE ions (high (H) atomic number (Z) and energy (E) ions). Protons are characterized by low linear energy transfer (LET) and HZE by high LET, as opposed to the low LET radiation (x-, β-, or γ-rays) to which humans are commonly exposed to on Earth [1,3,10,11,12,13]. HZE ions are positively charged and are the highly ionizing components of Galactic Cosmic Rays (GCRs) which originate outside of the solar system. The strong carcinogenic potential of HZE [14] has been a concern for many decades and has been considered the biggest risk factor for human health in missions beyond LEO [13,15]. In contrast, in LEO, where the ISS is located, the radiation dose from HZE is reduced due to the protection provided by the Earth’s magnetosphere, and is roughly equal to the dose from low-energy protons trapped in the inner radiation belt [13,16]. Most of the ISS orbit lies inside the radiation belts; the majority of the dose is from passages through the South Atlantic Anomaly, where the inner belt drops down to ISS altitudes. An additional risk for missions beyond LEO is from acute doses of high-energy protons from periodic solar particle events (SPEs) [14]. Data on the biological effects of space radiation has been obtained sporadically in true space conditions from samples flown in LEO (ISS, Space Shuttle, Mir Space Station, satellites carrying biology payloads). The majority of our data has been generated by simulating space radiation on Earth using specific HZE ions or protons at facilities such as the NASA Space Radiation Laboratory (NSRL) at Brookhaven National Laboratory (BNL). Simulated space radiation ground experiments have typically focused a lot more on HZE, reflecting the space radiation environment encountered in a long-duration beyond LEO mission [17]. To this day, a variety of results indicates that space radiation can increased cancer risk as well as contribute to diseases of the central nervous system (CNS), muscle degeneration, bone loss, and cardiovascular disease [10,15].

Radiation-induced late cardiovascular disease in astronauts is one increasingly prominent concern [7,18,19,20]. Although many experiments using low-LET radiation have been conducted to show damage to the microvasculature, HZE particle data are limited, and the implications for disease risk need more study. There have been even fewer ground-based studies of the synergistic effects of space radiation and microgravity, which more closely mimic the environment in space. Data from atomic bomb survivors, albeit with different types of radiation, on increased incidence of cardiovascular disease have shown long-lasting effects decades after radiation exposure impacting increased ischemic heart disease, hypertension, and stroke with doses as low as 2 Gy [21,22,23]. Animal and cell culture models have been used to simulate HZE particle and proton radiation and have shown that low doses of high-LET radiation can drastically affect the cardiovascular system [7,20,24,25,26,27,28]. Specifically, one experiment showed that mice exposed to 0.1 Gy of 150 MeV protons or 0.5 Gy of 600 MeV/n iron ions (^56^Fe) experienced large changes in cardiac DNA methylation with potential impacts including heart failure and cardiomyopathy [29]. ^56^Fe is the most predominant HZE particle in the GCRs, and is a commonly used surrogate for GCR HZE. Other experiments have shown that cells and animal models exposed to proton and HZE radiation increase DNA oxidation, myocardial fibrosis, endothelial dysfunction, and vascular damage [24,27,28,30]. Although these experiments and data have contributed to the overall knowledge of cardiovascular disease risk from space travel, there are many potential confounding factors besides space radiation (e.g., stress, hypoxia, microgravity). Therefore, our overall understanding of the contribution of space radiation in cardiovascular disease risk and their mechanisms remain an open question.

To further pursue the cardiovascular impact due to space radiation we utilized publicly available transcriptomic data that is available on NASA’s GeneLab platform (genelab.nasa.gov) [17] related to cardiovascular disease and the space environment. NASA’s GeneLab platform is a resource available to the public that has provided the first comprehensive omics database related to space. Due to the limited omics data available related to space radiation and cardiovascular disease, we analyzed two datasets from ground experiments designed to characterize the long-term impact following space irradiation (protons and ^56^Fe) on cardiomyocytes for five different time-points up to 28 days after irradiation and one spaceflight experiment studying the impact of human umbilical vein endothelial cells (HUVECs) cultivated in the space environment on the ISS for seven days. Two components that make up the cardiac tissue are cardiomyocytes and endothelial cells and both cell types are heavily involved with cardiac remodeling and regeneration [31]. Although there are more components to the cardiac tissue, from the datasets available from the GeneLab related to the cardiovascular system, we will only focus on these components. Using these datasets, we hypothesized that there are common long-term molecules and pathways modulating the cardiovascular system due to radiation effects during spaceflight. Using a systems biology approach, we were able to suggest a novel mechanism through the FYN pathway [32,33,34,35] that the cardiovascular system uses to reduce reactive oxygen species (ROS) levels when exposed to space radiation. Briefly, FYN is a member of the Src family of tyrosine kinases that has been previously shown to be heavily involved with the cardiovascular system through cardiomyocyte remodeling [36,37] and also leads to activation due to ROS [32,33,38,39]. In addition to FYN’s involvement with the cardiovascular system, it has also been reported to be a tumor suppressor [40,41], a key player in controlling immune receptor signaling status and inflammation [42], and has been shown to affect myelination of the central nervous system through crosstalk with Erk1/2 signaling [43]. For our findings, although the ROS levels are reduced through Fyn signaling, we show the potential of other pathways that will still contribute to increased cardiovascular risk in the space environment such as increased inflammation. In addition, we are able to suggest that HUVEC cells grown on the ISS in LEO show a molecular response that is in agreement with responses observed on Earth with simulated protons but with less commonality with heavier HZE ions, as one would expect from the predominance of protons found in LEO space radiation. This finding is critical as it highlights the relevance of simulated space radiation experiments conducted on Earth, as they are shown here as able to recapitulate what is observed in space. This work illustrates the power of unbiased omics analysis and how the GeneLab database has become an essential tool to better characterize the response of a life to space environment. More generally, these novel findings have potential to contribute to current cardiovascular risk models for spaceflight.

## 2. Results

### 2.1. Global Analysis of Simulated Space Radiation Ground Experiments Compared to Spaceflight Samples

To study the impact of space radiation on the cardiovascular system we utilized three GeneLab datasets (GLDS-52, -117, and -109). As described in detail in the methods, two datasets were focused on simulated space radiation studies performed on the ground. More specifically, male C57BL/6 mice received either an acute whole-body irradiation of 900 mGy proton at 1 GeV (GLDS-117) or 150 mGy ^56^Fe at 1 GeV/n (GLDS-109). Subsequently, cardiomyocytes were isolated and fixed from mice sacrificed at five different time-points post-irradiation. Unfortunately, 0 Gy control for the original study was acquired only at the first time-point (i.e., day 1). Although this is not the most optimal setup for these types of experiments, we will show that former datasets from GeneLab without the desired experimental setup can still produce meaningful results to guide space biology research. To test if simulated space radiation studies relate to the space environment, we also analyzed a spaceflight study (GLDS-52) which cultivated HUVECs on the ISS for 7 days before the cells were fixed. The HUVEC controls for the GLDS-52 datasets were cultured for the same length and methodology as the samples flown on the ISS, but were done on Earth [44]. Although all datasets utilized for this analysis were from microarray platforms, due to the incompatibility and batch effects that arise between different experiments, we had to analyze each dataset separately. We used either one-way ANOVA analysis for the simulated space radiation studies (all time-points were used to determine the significant genes) or *t*-test analysis with a *p*-value ≤ 0.05 for the spaceflight study. We produced the following statistically significant genes for each condition: for the proton study (GLDS-117) there were 2540 significantly regulated genes, for the ^56^Fe study (GLDS-109) there were 5554 significantly regulated genes, and for the spaceflight study (GLDS-52) there were 823 significantly regulated genes. We also checked the number of genes that would be produced with more stringent statistics using the adjusted Bonferroni method for a false discovery rate (FDR) of both ≤0.05 and 0.10. This approach produced the following number of genes for each condition: for the proton study (GLDS-117) there were 2 or 6 significantly regulated genes with either FDR ≤ 0.05 or 0.10, for the ^56^Fe study (GLDS-109) there were 7 or 12 significantly regulated genes with either FDR ≤ 0.05 or 0.10, and for the spaceflight study (GLDS-52) there were no significantly regulated genes with either FDR cutoff.

Note that our results do not agree with the original analysis published on the ground studies data [45]. Coleman et al. reported 5220 significant genes for ^56^Fe and 1651 significant genes for proton datasets using a *p*-value cutoff of 0.05 using one-way ANOVA analysis. Similarly, they reported 1538 significant genes using FDR < 0.10 with Benjamini–Hochberg correction. We were unable to reproduce the reported number of genes following the same procedure. Such discrepancy illustrates the importance of standard processing pipeline for analysis, as the protocol reported in Coleman et al. was not sufficient to allow reproducibility. One must note here that microarray analysis is highly sensitive in the way one normalizes data, with many available methods. For this reason, the Genelab team has established standardized processing workflows for all raw omics data (https://genelab-data.ndc.nasa.gov/genelab/projects). These pipelines were obtained as a consensus between more than 100 scientists who have joined the GeneLab Analysis Working Group (AWG) and covered transcriptomics (RNAseq and microarray), as well as metagenomics, epigenomics, proteomics, and metabolomics. The GeneLab team is in the process of documenting all these pipelines and making them available progressively online via the GeneLab toolshed (https://genelab-data.ndc.nasa.gov/glxy-sso-login/). For the two ground studies presented in this manuscript, the low number of replicates were incompatible with the stringent statistical method we recommend for GeneLab microarray data. Therefore, we could not utilize it here as it would not produce enough significant genes for a meta-analysis of this kind. We used instead a statistically acceptable but less stringent method (see Materials and Methods). We recommend the readers to go to the GeneLab toolshed for the latest recommended approach.

The global differences between all the time-points and conditions were assessed for each dataset (Figure 1). Using principal component analysis (PCA), we were able to distinguish global differences between the controls and the experimental groups (Figure 1A–C). Overall for both proton and ^56^Fe ion irradiation, the 0 Gy control samples were clustered separately from the rest of the irradiated samples (Figure 1A,B). It is interesting to note that the samples that produced the greatest separation were the cardiomyocytes isolated at the longest time-points (either 26 or 28 days) after irradiation. This is apparent in the principle component 2 (PC2) axis for protons (Figure 1A) and for ^56^Fe ions these points are clustering further apart in the negative PC1 axis from all other samples. This could be interpreted as space radiation (i.e., protons and ^56^Fe) producing long-lasting impact on the biology with a progressive response overtime. However, we cannot say for certain as there was no 0 Gy control done for the 26 day time-point. In contrast, when comparing spaceflight to ground controls for the HUVECs cultivated on the ISS (Figure 1C), we observed a much cleaner separation between the groups on the PCA plot.

When considering only the significantly regulated genes we see more refined separation between the different experimental groups (Figure 1D–F). For the significantly regulated genes, the 0 Gy controls in general were clustered independently from the rest of the irradiated samples with what appears to be a time dependence with the gene signature changing for each timepoint after the acute dose. There also was a distinct group of genes that were oppositely regulated for HUVECs flown in space compared to the ground controls. This data indicates that there is a distinct gene signature associated with space radiation and the impact of one acute dose can potentially have long-lasting effects on the cardiovascular system.

### 2.2. Dose Received by HUVECs Flown in Space

The data are from the European Space Agency-Integrated Experiment (ESA-SPHINX) experiment (10/31/10–11/6/10) [44]. The cells were launched to the ISS on a Russian Progress supply vehicle, cultured for 7 days in the KUBIK incubator on the European Space Agency (ESA) Columbus module, and then fixed. To be able to compare the dose received on the ISS to the simulated radiation ground studies, we obtained dose data from a detector located on the Columbus. (Data provided kindly by T. Berger from the German Aerospace Center (DLR)). The measured dose rate was almost constant throughout the experiment with an average of 220.43 µGy/day (in water) with a standard deviation of 1.90 µGy/day (in water) measured over the period of the experiment (Data provided kindly by T. Berger from the DLR and additional details can be found in the original publication [44]). The total dose was 1.544 mGy (in water) with the dose equivalent to 3.7 mSv, of which 58% was from high-energy protons and other GCR ions, and 42% from low-energy protons in the South Atlantic Anomaly (SAA). The actual dose to the samples was probably somewhat lower, as they were shielded by the incubator. Here it is important to make the distinction between particle energy and LET, which is essentially energy loss per unit path length and is proportional to the square of the charge. Linear energy transfer (LET) is a measure of how ionizing a particle is; because they are singly charged, both high- and low-energy protons are relatively low LET.

### 2.3. Common Pathways between Simulated Space Radiation and Spaceflight Samples

To fully understand the specific and long-lasting effects of space radiation on biology, we have to understand the general pathways that are being regulated. We first utilized gene set enrichment analysis (GSEA) [46] to determine significant pathways being regulated for each condition with an FDR ≤ 0.05. Using the Hallmarks (a more generalized pathway database, see Figure 2) and Reactome (Figure 3) gene sets for GSEA, we were able to identify key pathways that are commonly being regulated for all ground controls compared to the spaceflight samples or irradiated samples. We first observed that the majority of pathways for most time-points after radiation for the cardiomyocytes that were commonly regulated with the HUVEC spaceflight samples were downregulated (with the exception of one time-point at 1 day for samples irradiated with protons for some nodes) (Figure 2A). The time-points and samples were associated with each wedge in the node as displayed in the figure legend for Figure 2. Specifically, ROS pathways were surprisingly downregulated (indicating a decrease in ROS production) for the majority of the conditions, which suggests an immediate and persistent impact of space radiation on the cardiovascular system, which is also observed on flight samples as well. The other pathways that were downregulated have been shown from the literature to be connected to the ROS pathway. For example, fatty acid metabolism, which is one of the primary sources of energy for cardiac muscle, has a direct link to ROS formation and regulation through the mitochondria (Figure 2) [47]. It is interesting to note that it has been reported that the ROS production from the mitochondria is an important mechanism of how ROS impacts cardiac tissue [48]. Knocking out MYC has been shown to cause low levels of glycolysis and oxidative phosphorylation (Figure 2 and Figure 3) [49]. It is known that oxidative phosphorylation will produce ROS [50] and with the downregulation of MYC, this can potentially reduce oxidative phosphorylation levels which in turn will reduce ROS. Reactive oxygen species have also been shown to promote adipogenesis [51] and will be involved with heme metabolism (Figure 2) [52]. We also observed an overall downregulation of cell cycle pathways (Figure 3). We believe that this might be potentially due to the downregulation of ROS signaling, since it has been previously shown that ROS can heavily influence the cell cycle [53]. The downregulation of ROS seems to also lead to overall downregulation of cell cycle pathways (Figure 3). In general, ROS have been shown to be critically involved with the development of cardiovascular disease [52]. Through all these related downregulated pathways, we show that ROS potentially have long-lasting (Figure 2) effects directly linked to the space environment on the cardiovascular systems. On the other hand, surprisingly DNA repair hallmarks were downregulated for all irradiated samples at any time-point, whereas it was upregulated in space (Figure 2, top right panel). This result is unexpected as DNA repair is definitely expected early after exposure and suggests the cardiac response is quite different.

### 2.4. The Biological Pathways Elicited by Protons Have Stronger Overlap with Spaceflight Samples Than Those Elicited by ^56^Fe Ions

To test if one type of space radiation has a predominant effect on a spaceflight sample, one can compare the predicted pathways from one radiation condition alone with spaceflight samples. Figure 2C shows that the majority of pathways specifically in common with protons and spaceflight were upregulated, while Figure 2D shows the ^56^Fe component in common with spaceflight was downregulated. TGFβ signaling has been shown in the past to be a key player with regulating the overall system with proton irradiation (Figure 2) [11,54,55]. In order to further validate these results, we use another functional prediction tool, Ingenuity Pathway Analysis (IPA), to predict the involvement of specific upstream regulators, biofunctions, and canonical pathways in the various datasets (Figure 4A–C). In general, the overall pathways and functions overlap with the GSEA findings. For example, we observed in the upstream regulator predictions that MYC is downregulated for the majority for the pathways. We also observed that the following upstream regulators were predicted to activate or inhibit in the same directions as the GSEA predictions: p53, HIF1A, IL2, and IL6. The mTOR signaling canonical pathway also overlaps with the GSEA predictions. 

We were interested in determining how these overall functions from the ground experiments will group with the spaceflight samples. Using the activation *Z*-scores from the IPA results we were able to provide direct comparisons between all the samples, which we were not able to perform with the expression values due to the incompatibility of the microarray outputs. Using principal component analysis on *Z*-scores, we show that all cardiomyocyte samples irradiated with protons have biofunctions, upstream regulators and canonical pathways which cluster closely to the spaceflight samples (Figure 4D–F). In contrast, IPA functional predictions for cardiomyocytes from mice irradiated with ^56^Fe are nicely grouped together for three of the time-points (1 day, 3 days, and 14 days), while 7 days and 28 days seem to behave as outliers when compared to the other datasets. Overall, all time-points for ^56^Fe were clearly separated from the spaceflight and proton irradiated samples in the PC1 axis. Since the appropriate controls were not used for these datasets by the original investigators, we felt it appropriate to provide the potential that these two time-points may be outliers and a person may consider to remove these points from additional analysis. It is important to demonstrate how less than ideal GeneLab datasets can still have potential to provide meaningful results.

This result is also evident on Figure 4A–C, where most predicted IPA functions have negative *Z*-scores for 3 out of 5 ^56^Fe time-points, whereas all proton time-points and flight samples show positive *Z*-scores (red). These results indicate that the overall functional impact on samples caused by space radiation on the ISS are due to protons rather than HZE ions. In the past it has been difficult to determine which component of the ions in space will impact samples on the ISS. Previous models have predicted that due to SPEs and the protection of the magnetosphere around the Earth from HZE ions, astronauts should be mainly exposed to protons at LEO [2,10,13,15,16]. To our knowledge this is the first time suggesting from data utilized from the GeneLab that at LEO, protons will be the majority of ions that samples are exposed to. This data can provide a contribution to relating the biology being observed on the ISS to the correct type of irradiation that samples are exposed to.

### 2.5. Key Driving Genes Provide Direct Connection to ROS for Space Radiation Impact on the Cardiovascular System

To determine the key driving genes in an unbiased methodology we used an established systems biology approach that identifies the key genes involved in explaining the predicted biological functions changed by a specific treatment [54,55,56,57,58,59,60]. Our data has shown potential long-lasting impact of cardiomyocytes when comparing to the basal 1 day controls. Due to the original study design we did not have the optimal controls to compare to every time-point, but since there are relevant biological changes occurring when compared to the basal (1 day) ground controls, we hypothesis that there will be consistent long-lasting impact on cardiomyocytes isolated from mice irradiated with protons and ^56^Fe ion irradiation. In addition, this will also show how to creatively utilize less than ideal GeneLab datasets to generate meaningful biology to guide future research. For the key gene analysis, we pooled all irradiated time-points together to determine the common key driving genes for each type of radiation. Since we also observed that the 7 day and 28 day time-points for ^56^Fe vs. the control were acting as outliers (Figure 4D–F), we performed an additional key genes analysis excluding those time-points. This will only affect the key genes associated with ^56^Fe. This is an essential strategy to better identify the key genes specific to space radiation, which are able to relate back to the spaceflight data.

Utilizing the unbiased methodology described in the methods and further explained previously [54,55,56,57], we found utilizing all time-points nine persistent key genes for cardiomyocytes isolated from mice that received 150 mGy ^56^Fe and seven persistent key genes for cardiomyocytes isolated from mice that received 900 mGy protons. From the HUVECs flown on the ISS for seven days, we found nine key genes (Figure 5A). When excluding the outlier time-points for ^56^Fe (7 and 28 days), we obtain 85 key genes for cardiomyocytes isolated from mice that received 150 mGy ^56^Fe (Figure 5B). We were able to determine the connectivity of these key genes between the experiments through network analysis. This allowed us to find potential relationships with how the ground studies relate to the spaceflight experiments, and the central hubs or the most connected key genes that might be responsible for space radiation effects on the cardiovascular system. We found that the key/driving genes that are the central hubs are: *FYN*, *LCK*, *AKT1* being upregulated and *LYN* and *FOS* being downregulated with *FYN* being the overall central driver/hub for the cardiovascular response to space radiation (Figure 5A). It is worth noting the activation of *FYN* is a key event which prevents cardiac cell death and ROS production [35,36]. Interestingly it is known that *LCK*, *LYN*, and *FYN* are part of the SRC-family kinases (SFKs) [37,61,62,63] and have been associated with various aspects of cardiovascular disease [62,64,65,66,67]. In addition, *FYN* and *AKT1* are shown to be pivotal with angiogenic functions and the relation to cardiovascular disease [68,69]. Lastly, utilizing Z-scores through IPA we were able to show that these key genes provide a predicted inhibition of cell death of cardiomyocytes, congenital heart disease, and cardiac fibrillation (Figure 5). There also is a predicted inflammation of the organs which can cause negative health risk. Through an unbiased methodology we were able to show a link of key driving genes between ground and spaceflight studies that potential will have direct impact on the cardiovascular system. 

For the key genes analysis excluding the two outlier points (Figure 5B), since there were more key genes for ^56^Fe, additional predictions can be made for the functions related to cardiovascular system. For those key genes we show that in addition the functional impact described above, we also observe a predicted activation for damage to the heart. This can be attributed to potential damage caused by the key genes associated with ^56^Fe to promote damage to the heart tissue. In addition, we also observe three genes that overlap between the cardiomyocytes irradiation with protons and ^56^Fe: *CD4*, *CD3G*, and *LCK*. These genes are shown to be oppositely regulated between the irradiation conditions that can indicate potential different functional impact that can occur with the different ions. Since we are indicating that the majority of the ions that are affecting samples at LEO are protons (Figure 4D–F), for our interpretation of the overall data we will consider the upregulation of these three genes associated with proton irradiation to be the impact it will have on the samples.

### 2.6. Predicted Circulating miRNA Signature Associated with Space Radiation Cardiovascular Risk

We recently predicted a microRNA (miRNA) signature that was associated with microgravity and potential the space environment [54]. Utilizing a Cytoscape [70] plugin referred to as ClueGo [71], we were able to predict the top 30 miRNAs associated with the key genes (Figure 6). We believe that these miRNAs are associated with cardiovascular risk associated with space radiation. Interestingly, from an unbiased determination of the miRNAs, many of the predicted miRNAs have previously been associated with cardiovascular disease. For example 17 out of the 30 miRNAs predicted (miR-15a-5p, miR-16-5p, miR-29a-3p, miR-29b-3p, miR-101-3p, miR-125b-5p, miR-148a-3p, miR-153, miR-185-5p, miR-199a-3p, miR-221-3p, miR-302a, miR-520d-5p let-7a, let-7b, let-7d, and let-7f-1) have previously been reported to be either up- or downregulated in human heart disease and experimental models of heart failure [72]. Further, the let-7 family and miR-125b have been specifically shown to be involved with regulation of cardiomyocyte apoptosis [73]. Lastly, three of the miRNAs have overlap with our recently predicted miRNAs associated with the space environment which are: miR-125b-5p, miR-16-5p, and let-7a [54]. This can indicate from the original 13 miRNAs that comprised the systemic miRNA signature associated with spaceflight, that these three miRNAs are the components related to the cardiovascular system and space radiation.

## 3. Discussion

For long-term space travel, cardiovascular risk is a major concern to astronauts and is thought to arise from both microgravity effects and exposure to space radiation. Here we focused on how NASA’s GeneLab platform and datasets can be utilized to suggest the potential impact of space radiation on the cardiovascular system. Through an unbiased systems biology approach, we utilized three datasets (GLDS-52, -109, and -117) from NASA’s GeneLab platform that were related to space radiation and the cardiovascular system. Two of the datasets were from ground studies focused on cardiomyocytes that were isolated from mice after an acute whole-body dose of either 900 mGy of protons or 150 mGy of ^56^Fe [45]. The other study observed HUVECs for 7 days on the ISS [44]. As noted previously, the HUVECs received a total dose of 1.544 mGy, mostly from low LET protons. Through our unbiased analysis we were able to suggest: 1) that, consistent with the dose data, the majority of the radiation experienced on the ISS is potentially from protons (Figure 4D–F and Figure 2), a novel response mechanism of the cardiovascular system to prevent damage caused by space radiation through the activation of *FYN*.

By identifying the key genes in all three datasets, we proposed that *FYN* is the most connected gene and is therefore the central hub for this common biological response. This leads us to hypothesize that there is a feedback loop triggered by the oxidative stress induced by space radiation that upregulates *FYN* which in turn reduces ROS levels, and thus ROS pathways. Interestingly, this feedback loop is accompanied with a downregulation of the DNA repair pathway in the animals but not in the HUVECs flown in space, suggesting a response that is more targeted to the cardiovascular system in a full organism (Figure 2). By reducing ROS formation, cardiomyocyte and endothelial cells are less likely to die, leading to an overall protection of the cardiovascular system (Figure 7). More specifically, *FYN* was found to be upregulated within each group of the key genes associated with each dataset providing an overall predicted inhibition of ROS and cell death, while causing an activation of inflammation due to space radiation. This effect was suggested to be long-term since the key factors were from common genes being regulated the same direction up to 28 days after irradiation for the ground studies.

FYN, which is part of the SRC family kinases (which includes nine non-receptor tyrosine kinases discussed in the results), has been shown to be heavily involved with the cardiovascular system by decreasing ROS production and apoptosis [36]. Our analysis also agrees with what has been observed with the cardiovascular literature. It has been shown that knocking out *FYN* increases cardiac cell death and induces ROS production through NADPH Oxidase 4 (NOX4) signaling [36]. Others have shown *FYN* kinase inhibitors also inhibits nuclear factor erythroid 2–related factor 2 (NRF2) activity (which is an essential molecule for cardiac adaptive responses and protection) and enhances myocardial necrosis and death rate [39]. *NRF2* is also known to be essential in sensing ROS and regulating protection against damaging changes to the redox balance [74,75]. This mechanism has been discovered to be heavily regulated by *FYN* activation phosphorylating NRF2 which in turn triggers a defense response to ROS [75].

Space radiation has previously been shown to induce ROS in tissue and negatively impact the host [76,77]. What is novel about our finding is that the universal ROS impact that has classically been thought to impact all tissue does not seem to fit with cells involved with the cardiovascular system. Although ROS should be initially activated in cardiovascular tissue after exposure to space radiation, this mechanism through *FYN* attempts to protect the cardiovascular system. As discussed above, in a normal response with other types of external damage causing oxidative stress on the cardiovascular system, it has been shown that *FYN* will be activated and act as a negative feedback regulator to protect the cardiac tissue [36]. This mechanism has not been previously considered as an internal protective countermeasure to the effects of space radiation on cardiac tissue. Here we propose that there is a protective negative feedback loop that allows *FYN* to inhibit the ROS activation that is triggered by space radiation (Figure 7).

Although this seems to provide protection against a key player in space radiation damage to the cardiac tissue, other factors still seem to have a negative impact on the cardiovascular system that can increase risk of cardiovascular disease. To that end, we also observed a predicted activation of inflammation due to the key genes (Figure 4, Figure 5, and Figure 7). It is well known that inflammation in cardiac tissue has been associated with cardiovascular disease [78]. It has also been shown that reduction of ROS through antioxidants can potentially elevate chronic inflammation in cardiac tissue [78]. Our data suggests that although *FYN* might be acting as a natural antioxidant to reduce ROS effects, this mechanism might not be enough to reduce the inflammation caused by space radiation which has the potential to increase risk of cardiovascular disease.

From our previous studies, we have observed that miRNAs can be a key driver in response to the damage caused by the space environment [54]. We were able to predict the top 30 miRNAs that have potential to respond to space radiation in the cardiovascular system (Figure 6). Surprisingly, 17 out 30 miRNAs predicted have been shown to be associated with heart failure and play a role in cardiovascular disease [73]. Three out of 30 miRNAs also overlap with our previously identified universal miRNA signature that we have shown to systemically impact an organism due to a space environment. This preliminary result indicates that we have potential candidates for the specific component related to the cardiac tissue and space radiation. Currently, there is a lack of information for the role miRNAs in space radiation’s effect on the cardiovascular system. Our findings warrant further investigations to determine the extent to which miRNAs impact *FYN* and ROS due to space radiation.

We believe that the novel mechanism presented here for space radiation induced cardiovascular risk will impact the further development of space radiation risk models. We were able to directly link radiation ground studies to spaceflight and provide a rationale of how the cardiovascular system can help reduce ROS. We have shown through the power of NASA’s GeneLab platform that novel findings can be produced from existing data from former studies. By creatively combing several datasets we have potential indication of the specific space radiation component that astronauts are being exposed to at LEO on the ISS. It has been previously thought that protons comprise the majority of ions that astronauts are exposed to in LEO [16], but biological data have been lacking. Of course, further experiments have to be performed to determine whether these findings are true which can involve doing radiation ground experiments with HUVECs at 1.5 mGy (a comparable dose to the ISS for 7 days) and then once again comparing the gene expression changes to the dataset used here. The power of this type of analysis with GeneLab datasets allows for future guidance for space biology related experiments. In addition, we were able to uncover a novel mechanism underlying space radiation’s impact on the cardiovascular system. Although this study was done purely based on omics data, we believe that we have shown the power of using purely in silico methods to provide novel data. In addition, this manuscript demonstrates that GeneLab data can be utilized to give important hints to plan expensive space and accelerator experiments very carefully and not to be parsimonious when including the relevant controls and baselines. Further studies should be implemented to validate the findings reported here and potentially develop countermeasures for cardiovascular risk due spaceflight.

## 4. Materials and Methods

### 4.1. Datasets Utilized from the GeneLab Platform

All data used for this manuscript were obtained from GeneLab (genelab.nasa.gov). The following datasets were used: GLDS-52, -109, and -117. Spaceflight mission and ground radiation experimental details for each dataset such as, the handling of the cells, RNA extraction, and raw data pertaining to the microarray can be found in the GeneLab database. Briefly, we used two ground studies utilizing cardiomyocytes isolated from 8 to 9 months old male C57BL/6NT mice that have been whole-body irradiated with either one acute dose of 900 mGy protons at 1 GeV (GLDS-117) or 150 mGy of iron ions (^56^Fe) at 1 GeV/n (GLDS-109) and 5 time-points up to 28 days were collected after irradiation [45]. For protons, the following time-points were used for transcriptomic analysis: 1 day (*n* = 2), 3 days (*n* = 3), 5 days (*n* = 2), 12 days (*n* = 2), and 26 days (*n* = 2). For ^56^Fe, the following time-points were used for transcriptomic analysis: 1 day (*n* = 3), 3 days (*n* = 3), 7 days (*n* = 2), 14 days (*n* = 2), and 28 days (*n* = 2). For both experiments cardiomyocytes were also isolated from 0 Gy sham at the following time-points and data was pooled between all time-points for the controls: 1 day (*n* = 2) and 3 days (*n* = 1). All cardiomyocytes were isolated by microdissection from tissue dissections in order to remove other cell types. One spaceflight study was used that flew HUVECs to the ISS and the cells were exposed to space environment in vitro under standard conditions for 7 days before the cells were preserved in RNAlater [44]. The ground controls used for this study are 1 × *g* controls that were grown on Earth with identical conditions compared to the cells on the ISS.

### 4.2. Transcriptomic Analysis

The available transcriptomic data for tissues from GLDS-52, -109, and -117 datasets were previously performed on different versions of Affymetrix platforms. Due to the incompatibility and batch issues of processed data from these different platforms, all datasets were analyzed independently and the processed data was compared across all tissues. Datasets were background adjusted and quantile normalized using RMAExpress [79]. The probes from the pre-processed data were median collapsed using GenePattern [80]. All the data for each tissue/dataset was imported into MultiExperiment Viewer [81] and statistically significant genes was determined by ANOVA analysis between all conditions for GLDS-109 and -117, and a *p*-value ≤ 0.05 and *t*-test with a *p*-value for GLDS-52. The number of significant genes was used for the rest the analysis for the higher-order analysis.

Pathway analysis and subsequent predictions in each tissue were done using the statistically significant genes with a fold-change ≥1.2 (or ≤−1.2) comparing flight conditions versus ground controls or irradiated samples vs. unirradiated samples for ground experiments. Using a 1.2-fold-change cutoff allowed us to determine the impact of how major pathways were being regulated that would not be apparent with larger fold-change cutoffs. It was shown in the literature that larger fold-change values used for Omics data will significantly reduce useful biological data that becomes apparent when using lower fold-change cutoffs [82,83]. Ingenuity Pathway Analysis (IPA) software (Ingenuity Systems) was used to predict statistically significant activation or inhibition of upstream regulators, canonical pathways, biofunctions, and toxic functions using activation *Z*-score statistics (≥2, activated or ≤−2, inhibited) [84]. Gene set enrichment analysis (GSEA) was done using the C2, C5, and Hallmarks gene sets with a FDR ≤ 0.05 from the entire list of genes and additional leading edge analysis was performed as described by Subramanian et al. [46]. All heat maps and principal component analysis (PCA) plots were generated using packages available through R (pheatmap for heat maps and scatterplot3d for PCA plots).

### 4.3. Determination of Key Genes/Drivers

We used a previously established unbiased systems biology method to determine key genes/drivers [54,56,57,59,60] for each tissue. Briefly, the standard method we have developed in the past to determine key genes was done by determining the overlapping genes involved in the predictions made through IPA’s upstream regulators, canonical pathways, biofunctions, and GSEA gene sets (C2, C5, and Hallmarks gene sets). The common genes identified through these statistically significant predictions can be thought of as the central drivers for the experiment being studied, since the absence of the genes will make these predictions null. In some cases, these overlaps result in either no key genes or a very low number of genes. If this occurs, we adjust the pathway predictions involved with the gene overlaps. For the key genes determined here, we were forced to show the overlapping genes that were only involved with the GSEA Hallmarks Gene Set analysis with the IPA predictions.

To determine the key gene, which has the highest number of connections and can be thought of as the central hub for the set of key genes, we utilized IPA to define the total number of connections between all the key genes. Next, we plotted the genes using the radial plot tool which places the most connected gene in the center of the plot. Previous work involving similar statistically identified key genes provided experimental validation for this methodology using Western blots, qPCR, and other functional assays [56,60].

### 4.4. MicroRNA (miRNA) Predictions

Through the use of a Cytoscape [70] plug-in called ClueGo [71], we were able to predict miRNAs based on the key genes determined. This involved entering all key genes determined in ClueGo and allowing the software to determine the top 30 miRNAs that were significantly regulated and connected to the key genes. No directionality of the key genes was required for the results.

## Figures and Tables

**Figure 1 ijms-20-00661-f001:**
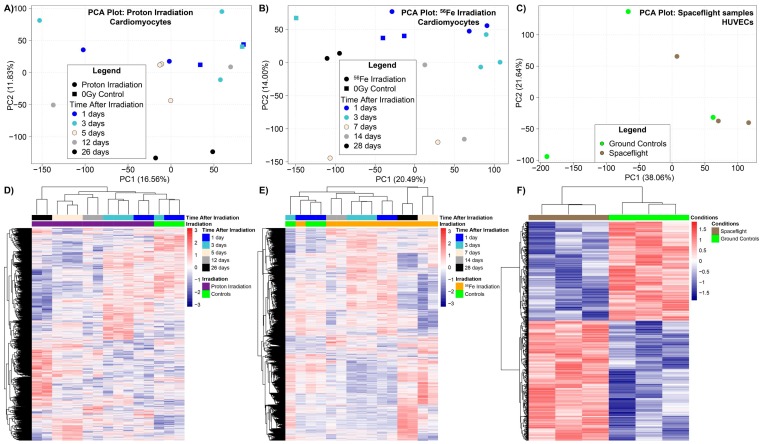
Global transcriptome analysis comparing experimental conditions to the controls for ground space radiation experiments and spaceflight experiment. The datasets analyzed were GeneLab Dataset (GLDS)-117 for cardiomyocytes isolated from C57BL/6 irradiated with whole body 900 mGy of proton at 1 GeV, GLDS-109 for cardiomyocytes isolated from C57BL/6 irradiated with whole body 150 mGy of ^56^Fe at 1 GeV/n, and GLDS-52 for Human Umbilical Vein Endothelial Cells (HUVECs) incubated on the International Space Station (ISS) for 7 days (estimated dose 1.5 mGy, see Section 2.2). (**A**–**C**) Principal component analysis (PCA) of all conditions for each dataset for genes. The percentage variance is shown for each principal component (PC) in parenthesis next to the PC axis. (**D**–**F**) Heat maps representing hierarchical clustering of significantly regulated genes by complete linkage and Euclidean distance calculation for datasets and conditions.

**Figure 2 ijms-20-00661-f002:**
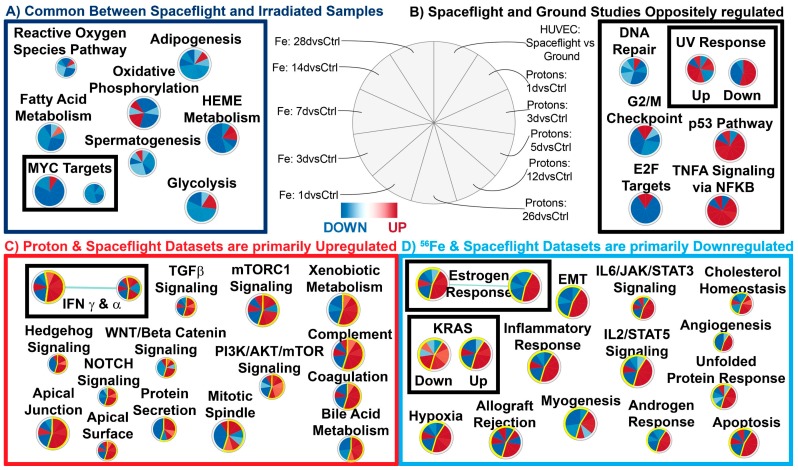
Gene set enrichment analysis (GSEA) emphasizing the hallmark gene sets commonly enriched in cardiomyocytes of irradiated animals or HUVECs flown in space. (**A**) Common Hallmark gene sets enriched in spaceflight samples and irradiated samples. The majority of these datasets are downregulated. (**B**) Hallmark gene sets that are oppositely regulated between all irradiated samples and the spaceflight samples. (**C**) Hallmark gene sets commonly enriched in cardiomyocytes of animals irradiated with 1 GeV/n proton or HUVECs flown in space. Note, most of them are upregulated. (**D**) Hallmark gene sets commonly enriched in cardiomyocytes of animals irradiated with 1 GeV/n ^56^Fe or HUVECs flown in space. Note, most of them are downregulated for ^56^Fe and spaceflight samples. Note: all samples were comparing experimental conditions with the respective controls. For figures (**C**,**D**), the common wedges are highlighted in yellow outline. The significant gene sets were determined with false discovery rate (FDR) ≤ 0.05. The legend in the middle of the figure displays what each component/wedge of the nodes represent. Each node contains 11 wedges for each condition and the color of each wedge indicates if the gene set is downregulated (blue) or upregulated (red). The shade of the color indicates the degree of regulation. Each node represents one gene set and the size of the node indicates the number of genes involved with the predictions. Pathways with more than one node are grouped together in the black boxes under each major category.

**Figure 3 ijms-20-00661-f003:**
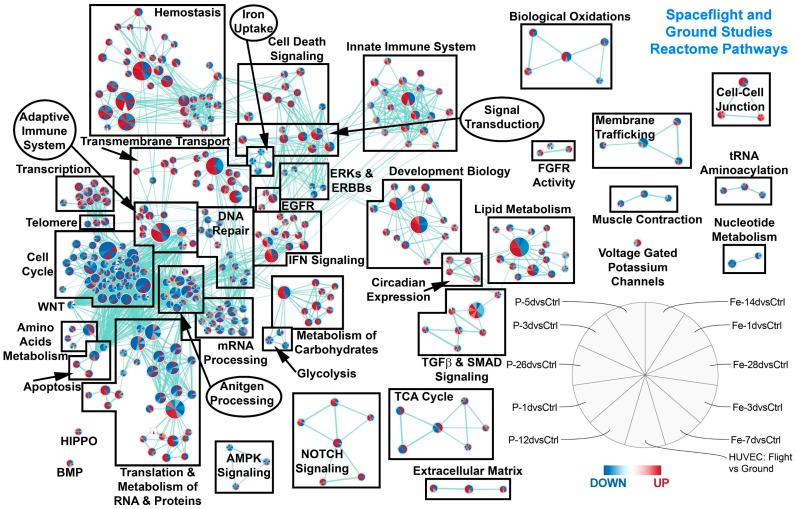
Gene set enrichment analysis (GSEA) with the Reactome gene sets comparing different time-points after irradiation with either protons or ^56^Fe with the HUVECs flown in the space. All samples were comparing experimental conditions with the respective controls. The significant gene sets were determined with FDR ≤ 0.05. Each node represents one gene set and the size of the node indicates the number of genes involved with the predictions. Each node contains 11 wedges for each condition and the color of each wedge indicates if the gene set is downregulated (blue) or upregulated (red). The shade of the color indicates degree of regulation.

**Figure 4 ijms-20-00661-f004:**
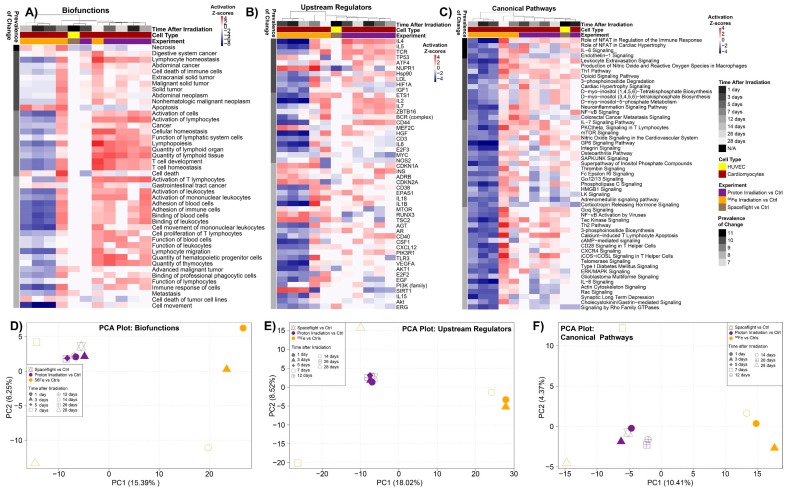
Predicted functions and upstream regulators affected by space radiation for HUVECs exposed to the space environment and cardiomyocytes from mice exposed to protons and ^56^Fe ion irradiation with multiple time-points after irradiation. The predicted statistically significant biofunctions (**A**), upstream regulators (**B**), and canonical pathways (**C**), determined through Ingenuity Pathway Analysis (IPA) from data for each individual dataset using activation *Z*-score statistics. Heat map representation of the activation *Z*-score values (red = positive activation *Z*-score for activation and blue = negative activation *Z*-score for inhibition) with hierarchical clustering by complete linkage and Euclidean distance calculation for datasets and conditions were used to display the data. The prevalence of change (or % of dataset) on the left side of the heat maps represents how common that factor is throughout all datasets/tissues with the darkest color representing factors with the highest degree in common. Time after irradiation, cell type, and experiment is color coded on the top of the heat maps. For all heat maps we are only showing the prevalence of change for the functions associated with the top 4 (i.e., a prevalence of change score ≥ 7). Global clustering of the biofunctions (**D**), upstream regulators (**E**), and canonical pathways (**F**) for all samples are shown by principal component analysis (PCA). For the PCA plots we used all functions for each datapoint regardless of the prevalence of change score.

**Figure 5 ijms-20-00661-f005:**
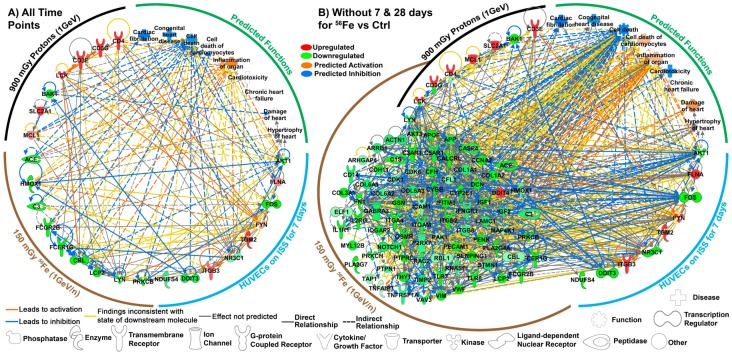
The key driving genes associated with space radiation and the predicted functions that these genes will affect. A circular network created with IPA is utilized connecting all the key genes from each experimental condition (i.e., common key genes for all time-points for both cardiomyocytes from mice irradiated with either 150 mGy ^56^Fe or 900 mGy protons compared to 0 Gy controls and HUVECs flown in on the ISS for 7 days compared to the ground control). The color of the gene represents whether the gene is upregulated (red) or downregulated (green) with the shade signifying the degree of regulation. (**A**) The network of key genes done with analysis using all time-points and samples. (**B**) The network of key genes for analysis done with all time-points except for 7 and 28 days for ^56^Fe vs Ctrl. For this analysis there were three key genes that overlapped between the cardiomyocytes irradiated with protons and ^56^Fe (*CD4*, *CD3G*, and *LCK*). These were oppositely regulated between the two irradiation conditions and the colors in the top half indicate the gene expression for protons and the bottom indicates the gene expression for ^56^Fe. The different line colors represent the predicted effect of each gene on each other. Predicted functional impact of the key genes were determined using IPA which allowed us to predict whether cardiac related functions were either inhibited (blue) or activated (orange). The identification of the symbols for each gene is shown on the bottom of the figure.

**Figure 6 ijms-20-00661-f006:**
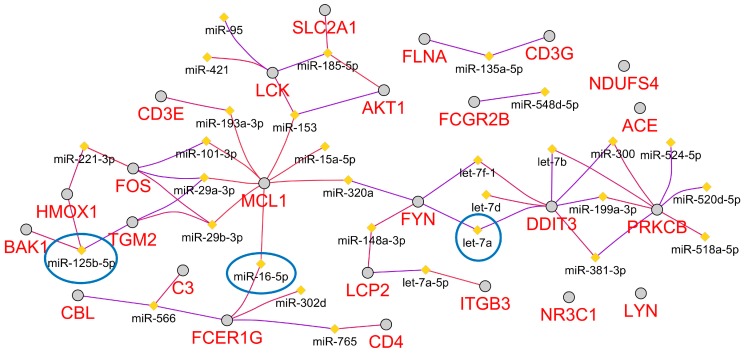
The top 30 miRNAs associated with all the key genes determined by Cytoscape [70] plugin referred to as ClueGo [71].

**Figure 7 ijms-20-00661-f007:**
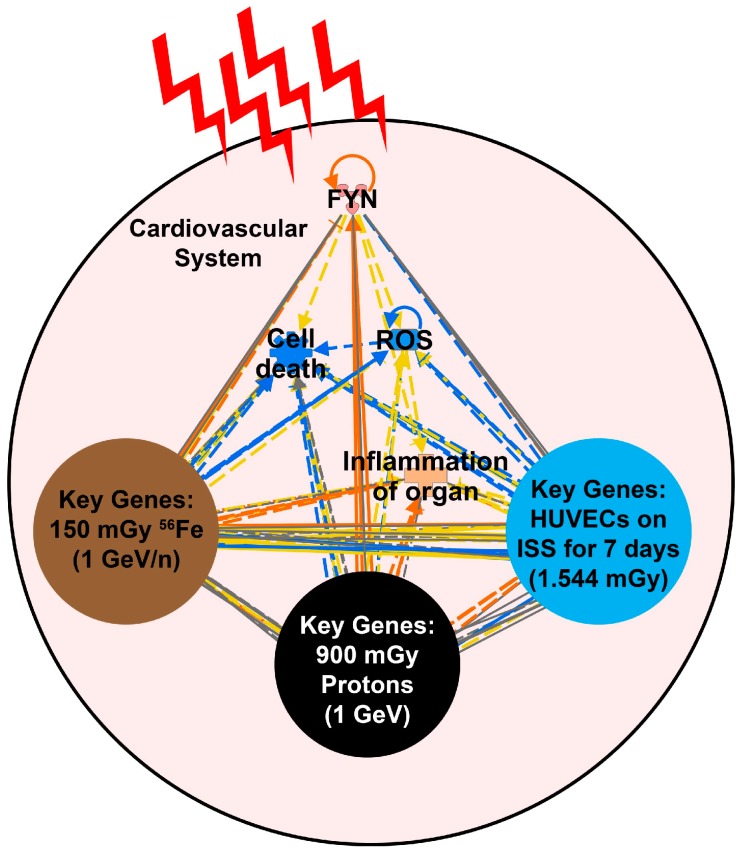
A schematic of the mechanism involved with the space radiation impact on the cardiovascular system. We determined that through the upregulation of *FYN* (the determined key/central driver involved with space radiation), ROS is downregulated to protect against the upregulation due to space radiation. Although *FYN* is predicted to be the central player, the other key genes involved from each dataset (Figure 5) will contribute to the reduction in ROS and cell death. In addition, we also observe potential increased cardiovascular health risk due to increased predictions of inflammation.

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
