# Peer review of "GeneLab Database Analyses Suggest Long-Term Impact of Space Radiation on the Cardiovascular System by the Activation of FYN Through Reactive Oxygen Species"

_ijms, 2019, doi:10.3390/ijms20030661_

Reviewer 1 Report

The comments to the revised version were addressed and the editor intervened in order to reduce the strength of the statements in question (long-term effects, biological evidence for protons as major components of the ISS radiation field).

I agree that interesting research questions and endpoints can be generated by GeneLab database analyses, but these analyses cannot improve the quality of the original data.

The paper thereby gives important hints to plan such expensive space and accelerator experiments very carefully and not to be parsimonious when including the relevant controls and baselines. Maybe this can also be a conclusion of this publication.

 Author Response

We thank the reviewer for their insightful comments and have added the following sentence to the end of the conclusion as suggested:

 “In addition, this manuscript demonstrates that GeneLab data can be utilized to give important hints to plan expensive space and accelerator experiments very carefully and not to be parsimonious when including the relevant controls and baselines.”

We have kept the track changes on so the reviewer can easily see where this addition is included in the manuscript.

Reviewer 2 Report

The authors have addressed my comments. 

Minor comments:

Line 83-84 - the sentence was removed during revision, but a portion of the sentence still remains. Remove, "More recently, data from astronauts involved with NASA's"

Line 102 - add "system" after "cardiovascular" 

Author Response

We thank the reviewer for their comments.

We have addressed their comments with the following:

Line 83-84 - the sentence was removed during revision, but a portion of the sentence still remains. Remove, "More recently, data from astronauts involved with NASA's"

We have removed this sentence.

Line 102 - add "system" after "cardiovascular" 

We have added “system” after “cardiovascular” in that sentence.

We have kept the track changes on in the manuscript so the reviewer can easily locate these changes.

This manuscript is a resubmission of an earlier submission. The following is a list of the peer review reports and author responses from that submission.

Round  1

Reviewer 1 Report

The current manuscript details an analysis of the flight and ground-based omics data concerning the effects of space radiation and/or microgravity (spaceflight sample) on health of cardiovascular cells (e.g., HUVEC). 

The analysis describes novel and important data and demonstrates that ground-based space radiation studies overlap with apparent radiation-induced changes in HUVEC during spaceflight. The analysis sheds light on the possible mechanisms underlying these effects, such as changes in the FYN pathway. 

Overall, the manuscript is well-written, provides novel data concerning space radiation, and demonstrates that ground-based animal studies examining the effects of space radiation on the cardiovascular system have high relevance to the effects of spaceflight in humans. 

This reviewer is requesting minor additions to the introduction/discussion and minor changes to some of the figures to increase readability.

Introduction/discussion - in one of these sections it would help the reader if the authors could include description of what FYN is. They provide minimal description in the results section (e.g., src-kinase). Including the role of FYN in normal cardiovascular health would be helpful, in addition to the roles this kinase has in any other tissues that could be negatively impacted by space radiation (e.g., neural tissues).  

Figure 1. For panels D-F, the variations in grey bars are difficult to discern when looking at the bars above the graph. It might be better for the reader if the authors used separate colors for the different time points. It is most problematic for the two earliest time points on panel E. Also on panel E, there is a row that is labeled with orange of 56Fe intermixed with the control bars. The authors should double-check this labeling to make sure it is correct. 

Author Response

December 13th, 2018

Dear Reviewer 1,

 We thank you for providing thoughtful comments for our manuscript. I have provided the revised manuscript with track changes so the editors and reviewers can easily view the requested changes. We have also provided revisions for the following figures based on the comments provided by the reviewers: Figure 01, Figure 02, Figure 05, Figure 07, and the Graphical Abstract. In addition, we have now included the Supplemental Figure as a figure in the manuscript based on Reviewer #2’s comments. This figure is now labeled as Figure 03 and all figure labels have been adjusted accordingly throughout the manuscript.

 Below we have provided a detailed response to your comments and concerns. We have provided both the reviewer’s comments and our responses to each comment in red text. We believe we have adequately responded to the reviewers comments and look forward to hearing back from you.

 Afshin Beheshti, PhD

 Reviewer #1

 Comments and Suggestions for Authors

The current manuscript details an analysis of the flight and ground-based omics data concerning the effects of space radiation and/or microgravity (spaceflight sample) on health of cardiovascular cells (e.g., HUVEC). 

 The analysis describes novel and important data and demonstrates that ground-based space radiation studies overlap with apparent radiation-induced changes in HUVEC during spaceflight. The analysis sheds light on the possible mechanisms underlying these effects, such as changes in the FYN pathway. 

Overall, the manuscript is well-written, provides novel data concerning space radiation, and demonstrates that ground-based animal studies examining the effects of space radiation on the cardiovascular system have high relevance to the effects of spaceflight in humans. 

 We thank the reviewer for their thoughtful comments and have tried to address the comments below.

 This reviewer is requesting minor additions to the introduction/discussion and minor changes to some of the figures to increase readability.

 Introduction/discussion - in one of these sections it would help the reader if the authors could include description of what FYN is. They provide minimal description in the results section (e.g., src-kinase). Including the role of FYN in normal cardiovascular health would be helpful, in addition to the roles this kinase has in any other tissues that could be negatively impacted by space radiation (e.g., neural tissues).  

 We have added the following starting at line 120 on p. 3 to the introduction to provide a little more background on FYN:

“Briefly, FYN is a member of the Src family of tyrosine kinases that has been previously shown to be heavily involved with the cardiovascular system through cardiomyocyte remodeling [37,38] and also leads to activation due to ROS [33,34,39,40]. In addition to FYN’s involved with the cardiovascular system, it has also been reported to be a tumor suppressor [41,42], a key player in controlling immune receptor signaling status and inflammation [43], and has been shown to affect myelination of the central nervous system through crosstalk with Erk1/2 signaling [44].”

 Figure 1. For panels D-F, the variations in grey bars are difficult to discern when looking at the bars above the graph. It might be better for the reader if the authors used separate colors for the different time points. It is most problematic for the two earliest time points on panel E. Also on panel E, there is a row that is labeled with orange of 56Fe intermixed with the control bars. The authors should double-check this labeling to make sure it is correct.

 We have adjusted the color scale for figure 1 to provide better clarity for the time points. In addition, in panel E the 56Fe point that the reviewer has pointed is actually real. That specific sample is clustering more closely to the control values for that time point.

 Reviewer 2 Report

The authors performed a reanalysis and comparison of microarray data from ground-based animal experiments at a heavy ion accelerator and a cell culture experiment in the KUBIK incubator on the International Space Station (ISS). Cardiomyocytes were isolated after proton and iron ion irradiation of mice. Endothelial cells were cultured for seven days on the ISS.

Gene expression after irradiation (early and late time points) and after incubation on ISS was compared. Key pathways, genes and miRNAs were identified by bioinformatics.

 Major comments:

The comparison of ground based and inflight experimental data is a very useful approach, especially as spaceflight and accelerator-exposed samples are expensive to be produced and rare opportunities. However, unfortunately, the original data lack important controls, and the differences in irradiation dose make conclusions from this study rather shaky.

 Title: Please try to avoid abbreviations in the title

 Introduction:

p. 1 line 40      “low LET radiation (x-,a-, β-, or g-rays)” => a-radiation can be high LET, e.g. ~ 150 keV/µm – energy 0.5 MeV/n

 p. 1 line 41-42 “HZE ions are positively charged and are the primary components of Galactic Cosmic Rays (GCR) which originate outside of the solar system.” => Quantitatively, protons are the “primary components”, the HZE particles make up only ~ 1 %. From the point of view of biological effectiveness, they can be seen as “primary components”. Please clarify.

 p. 2 line 77-80 “Specifically, it was shown that the mortality rate due to cardiovascular disease for astronauts who flew to the Moon was approximately 20% higher than that in the normal US population and that exposure to space radiation causes long lasting vascular endothelial dysfunction [30].” => This article has provoked a lot of critics and it was shown that the conclusions are not supported by the results presented in the paper. It should not be cited in such an uncritical way.

 Cucinotta FA, Hamada N, Little MP. No evidence for an increase in circulatory disease mortality in astronauts following space radiation exposures. Life Sci Space Res (Amst). 2016 Aug;10:53-6. doi: 10.1016/j.lssr.2016.08.002.

 Results:

 GLDS-117 and GLDS-109: “Unfortunately, 0 Gy control were acquired only at the first time point (i.e. day 1)” => This is a major problem for the whole study and makes the results for the time-points after day 1 questionable.

 p. 5 line 190:   In addition to the energy dose, the dose equivalent should be available, or the average Q factor which would give a hint about the biological effectiveness of the radiation field.

 Figure 2: The order of the samples in the GSEA plot is quite confusing, why not order them according to the incubation time?

The meaning of the inset boxes (myc targets, UV Response, Estrogen Response, KRAS) is unclear.

 p. 8 line 284-285: “These results indicate that the overall functional impact of space radiation on the ISS on samples is due to protons rather than HZE ions.”

p. 10 line 381-382: “1) that, consistent with the dose data, the majority of the radiation experienced on the ISS is from protons”

p. 12 Lines 450-452: “By creatively combing several datasets we were able to isolate for the first time the specific space radiation component that astronauts are being exposed to at LEO on the ISS.”

=> Regarding the difference in total dose (1.5 mGy accumulated on the ISS within 7 days, and 900 mGy protons in the accelerator experiment), I don’t think that such a conclusion is possible. Here, a comparison of the gene expression profile after incubation on the ISS and exposure on ground to 1.5 mGy protons at an accelerator would be necessary. The influence of other spaceflight factors including microgravity and the experimental conditions on the ISS might play a much higher role than the tiny radiation dose. Also, different cell types are compared by this approach (cardiomyocytes and endothelial cells). These many influencing factors do not allow the conclusions that are cited above. Especially as the results are suggested to be considered in risk models, high care has to be taken that the data support the conclusions.

 Rp. 8 lines 296-297: “Since our data has shown consistent long-lasting impact on cardiomyocytes isolated from mice irradiated with protons and 56Fe irradiation” => As the controls for the late time-points are missing, no conclusions about long-lasting effects are possible. This is especially true as the 7 and 28 days results are outliers.

 p. 13 lines 486-487: “Pathway analysis and subsequent predictions in each tissue were done using the statistically significant genes with a fold-change ≥1.2 (or ≤-1.2)”. Even if the changes were statistically significant, what is the biological relevance of such small gene expression changes?

 Results:

p. 3 line 121 GLDS-52 => here, short information on the controls (inflight 1 x g centrifuge, ground controls …) is missing.

 p. 3 line 126-127 “for the proton study (GLDS-117) there were 2540 significantly regulated genes” => it is unclear for which time point this applies.

 p. 3 line 126-127         “for the 56Fe study (GLDS-109) there were 5554 significantly regulated genes” => it is unclear for which time point this applies.

 p. 4 line 171    “estimated dose 1.544 mGy” => To my opinion, there are too many digits for an estimation only.

 Supplemental Figure:

I think the figure should be included in the manuscript as it does not take much space and is discussed in detail and it is inconvenient for the reader to have to download it separately.

 Materials and Methods:

 This part lacks some important information:

 p. 12 line 471 “cardiomyocytes were isolated” => Did the researchers take the cardiomyocytes in culture? If, for which time period? Or was it a tissue dissection in order to remove other cell types?

 p. 12 line 473-475 It is unclear which controls (1 x g centrifuge on ISS ?, ground controls?) were used in this spaceflight experiment.

 Minor comments:

p. 1 line 16                  “male C57BL/6” => “male C57BL/6 mice”

 p. 1 line 16      “90cGy” => cGy is not a SI unit and should not be used. Please use Gy or mGy (check the whole manuscript).

 p. 1 line 19      “FYN” => please explain all abbreviations at their first appearance in the abstract and in the text.

 p. 1 line 23      “Protons” => “protons”

 p. 1 line 34                  “LEO (Low Earth Orbit)” => “Low Earth Orbit (LEO)”

 p. 1 line 38                  “earth” => “Earth”

 p. 2 line 53                  “earth” => “Earth”

 p. 3 line 104                “earth” => “Earth”

 p. 3 line 107                “earth” => “Earth”

 p. 5 line 184:   “2.2. .Dose Received by HUVECs Flown in Space” => “2.2. Dose Received by HUVECs Flown in Space”

 p. 5 line 206    “reactive oxygen species (ROS)” => “ROS”

 p. 5 line 208    “on the cardiovascular systems” => “on the cardiovascular system”

 p. 5 line 213    “ROS has also been shown” => “ROS have also been shown”

 p. 5 line 215    “ROS has been shown” => “ROS have been shown”

 p. 5 line 217:   “that ROS is having long lasting effects” => “that ROS are having long lasting effects”

 p. 5 line 218:   “DNA repair Hallmarks” => “DNA repair hallmarks”

 p. 7 line 245: “2.4. The Biological Pathways Elicited by Protons have stronger overlap with Spaceflight Samples than 56Fe” => “2.4. The Biological Pathways Elicited by Protons have stronger overlap with Spaceflight Samples than those elicited by 56Fe ions”

 p. 7 line 246 “one type of space irradiation” => “one type of space radiation”

 p. 8 line 297 “56Fe irradiation” => “56Fe ion irradiation” (check the whole manuscript)

 p. 8 line 305 “time points” => time-points (spelling varies within the manuscript, please check)

 p. 8 line 317 “Interestingly it known” => “Interestingly it is known”

 p. 9 line 325 “leads to acitvation” => “leads to activation”

 p. 9 line 336 “the two irradiated conditions” => “the two irradiation conditions”

 p. 9 line 336 “the colors in the top half indicates” => “the colors in the top half indicate”

 p. 12 line 466 “1GeV” => “1 GeV”

 p. 14 line 539 “Hze” => “HZE”

 p. 14 line 542 “mars” => “Mars”

 p. 15 line 568 “Nasa genelab” => “NASA GeneLab”

 p. 15 line 573 “(vegf)” => “(VEGF)”

 p. 15 line 588 “si” => “Si”

 p. 15 line 590 “let” => “LET”

 p. 15 line 590 “fe” => “Fe”

 p. 15 line 593 “apolipoprotein e” => “apolipoprotein E”

 p. 15 line 610 “fyn” => “FYN”

 p. 16 line 619 “nrf2” => “Nrf2”

 p. 16 line 632 “acetyl-coa” => “acetyl-coA”

 p. 16 line 638 “microrna” => “microRNA”

 p. 16 line 638 “tgf-beta1” => “TGF-beta1”

 p. 16 line 656 “t-cell” => “T-cell”

 p. 16 line 658 “fyn” => “FYN”

 p. 16 line 660 “src-family kinases, lck and fyn, in t-cell” => “SRC-family kinases, LCK and FYN, in T-cell”

 p. 17 line 678 “Elevating cxcr7 improves angiogenic function of epcs via akt/gsk-3beta/fyn-mediated nrf2 activation” => “Elevating CXCR7 improves angiogenic function of EPCs via Akt/GSK-3beta/FYN-mediated Nrf2 activation”

 p. 17 line 681 “nrf2 suppression via modulation of akt/gsk3beta/fyn kinase” => “Nrf2 suppression via modulation of Akt/GSK-3beta/FYN kinase”

 p. 17 line 689 “Micrornas” => “MicroRNAs”

 p. 17 line 696 “nrf2” => “Nrf2”

 Reference list: check journal abbreviations / names

Author Response

December 13th, 2018

 Re: addressing reviewers comments

 Dear Reviewer 2,

 We thank you for providing thoughtful comments for our manuscript. I have provided the revised manuscript with track changes so the editors and reviewers can easily view the requested changes. We have also provided revisions for the following figures based on the comments provided by the reviewers: Figure 01, Figure 02, Figure 05, Figure 07, and the Graphical Abstract. In addition, we have now included the Supplemental Figure as a figure in the manuscript based on Reviewer #2’s comments. This figure is now labeled as Figure 03 and all figure labels have been adjusted accordingly throughout the manuscript.

 Below we have provided a detailed response to your comments and concerns. We have provided both the reviewer’s comments and our responses to each comment in red text. We believe we have adequately responded to the reviewers comments and look forward to hearing back from you.

 Afshin Beheshti, PhD

 Reviewer #2

 Comments and Suggestions for Authors

The authors performed a reanalysis and comparison of microarray data from ground-based animal experiments at a heavy ion accelerator and a cell culture experiment in the KUBIK incubator on the International Space Station (ISS). Cardiomyocytes were isolated after proton and iron ion irradiation of mice. Endothelial cells were cultured for seven days on the ISS.

 Gene expression after irradiation (early and late time points) and after incubation on ISS was compared. Key pathways, genes and miRNAs were identified by bioinformatics.

 Major comments:

The comparison of ground based and inflight experimental data is a very useful approach, especially as spaceflight and accelerator-exposed samples are expensive to be produced and rare opportunities. However, unfortunately, the original data lack important controls, and the differences in irradiation dose make conclusions from this study rather shaky.

Unfortunately, the setup of the original experiment is out of our control. The point of this study is to show how GeneLab datasets can be used to generate novel and interesting hypothesis to guide future space biology experiments. As stated in the manuscript, based on the controls provided by the dataset, we are still able to find relevant biology that agreed with what is known in the literature that can be applied to space radiation effects for cardiovascular disease. We have indicated this in the manuscript (page 4 line 171). We also have added the following text (page 3 line 125) to further explain this point:

“Although this is not the most optimal setup for these types of experiments, we will show that former datasets from GeneLab without the desired experimental setup can still produce meaningful results to guide space biology research.”

 Title: Please try to avoid abbreviations in the title

 In title we used the gene name for FYN which is common practice when using gene names in the title. We have tried to avoid using any abbreviations other than this.

 Introduction:

p. 1 line 40      “low LET radiation (x-,a-, β-, or g-rays)” => a-radiation can be high LET, e.g. ~ 150 keV/µm – energy 0.5 MeV/n

 Thank you for pointing this out. What the reviewer has brought is correct and we have removed a-radiation from the low LET radiation list.

 p. 1 line 41-42 “HZE ions are positively charged and are the primary components of Galactic Cosmic Rays (GCR) which originate outside of the solar system.” => Quantitatively, protons are the “primary components”, the HZE particles make up only ~ 1 %. From the point of view of biological effectiveness, they can be seen as “primary components”. Please clarify.

 We have changed the wording for this from “primary components” to “highly ionizing”.

 p. 2 line 77-80 “Specifically, it was shown that the mortality rate due to cardiovascular disease for astronauts who flew to the Moon was approximately 20% higher than that in the normal US population and that exposure to space radiation causes long lasting vascular endothelial dysfunction [30].” => This article has provoked a lot of critics and it was shown that the conclusions are not supported by the results presented in the paper. It should not be cited in such an uncritical way.

Cucinotta FA, Hamada N, Little MP. No evidence for an increase in circulatory disease mortality in astronauts following space radiation exposures. Life Sci Space Res (Amst). 2016 Aug;10:53-6. doi: 10.1016/j.lssr.2016.08.002.

 We thank the reviewer to bringing this to our attention and we have removed this reference and sentence from our manuscript.

 Results:

 GLDS-117 and GLDS-109: “Unfortunately, 0 Gy control were acquired only at the first time point (i.e. day 1)” => This is a major problem for the whole study and makes the results for the time-points after day 1 questionable.

 As stated above we agree that the original study we used from GeneLab could have done a better job with their controls. That said

 p. 5 line 190:   In addition to the energy dose, the dose equivalent should be available, or the average Q factor which would give a hint about the biological effectiveness of the radiation field.

 We have added the following on line 243 after “…the total dose was 1.544 mGy (in water)…” to address this comment:  “…with the  dose equivalent being 3.7 mSv,…”

 Figure 2: The order of the samples in the GSEA plot is quite confusing, why not order them according to the incubation time? The meaning of the inset boxes (myc targets, UV Response, Estrogen Response, KRAS) is unclear.

 We have reordered the GSEA plot according to incubation times. The inset boxes are pathways that have more than one node associated with it. In the figure legend we have added the following sentence to avoid confusion (page 6 line 314):

“Pathways with more than one node are grouped together in the black boxes under each major category.”

 p. 8 line 284-285: “These results indicate that the overall functional impact of space radiation on the ISS on samples is due to protons rather than HZE ions.”

p. 10 line 381-382: “1) that, consistent with the dose data, the majority of the radiation experienced on the ISS is from protons”

p. 12 Lines 450-452: “By creatively combing several datasets we were able to isolate for the first time the specific space radiation component that astronauts are being exposed to at LEO on the ISS.”

=> Regarding the difference in total dose (1.5 mGy accumulated on the ISS within 7 days, and 900 mGy protons in the accelerator experiment), I don’t think that such a conclusion is possible. Here, a comparison of the gene expression profile after incubation on the ISS and exposure on ground to 1.5 mGy protons at an accelerator would be necessary. The influence of other spaceflight factors including microgravity and the experimental conditions on the ISS might play a much higher role than the tiny radiation dose. Also, different cell types are compared by this approach (cardiomyocytes and endothelial cells). These many influencing factors do not allow the conclusions that are cited above. Especially as the results are suggested to be considered in risk models, high care has to be taken that the data support the conclusions.

 We agree with the reviewer that further experiments will have to be done fully validate these findings. We believe that the power of this type of analysis with publicly available datasets from GeneLab, allows the public to potentially determine the best future experiments to perform. As stated in the manuscript we believe, since the microgravity component is not present with the mice ground experiments it provides a nice way to only dissect the radiation component when comparing to spaceflight studies. We believe that these results can provide some evidence of the specific space radiation component experienced at LEO. That said, we have rephrased our statements regarding this issue to provide the “potential” that protons are the majority of the type of radiation experiences at LEO. Here are the rephrased sentences:

 “1) that, consistent with the dose data, the majority of the radiation experienced on the ISS is potentially from protons”

 “By creatively combing several datasets we have potential indication for the first time of the specific space radiation component that astronauts are being exposed to at LEO on the ISS. It has been previously thought that protons comprise the majority of ions that astronauts are exposed to in LEO [15], but biological data have been lacking. Of course, further experiments have to be performed to determine whether these findings are true, but the power of this type of analysis with GeneLab datasets allows for future guidance for space biology related experiments.”

 p. 8 lines 296-297: “Since our data has shown consistent long-lasting impact on cardiomyocytes isolated from mice irradiated with protons and 56Fe irradiation” => As the controls for the late time-points are missing, no conclusions about long-lasting effects are possible. This is especially true as the 7 and 28 days results are outliers.

 Although the original study design isn’t optimal, we believe that comparing to the original basal controls (1 day control group) will at least allow us to determine how the biology changes when compared to the original start of the experiment. We have rephrased this part to indicate this:

 “Our data has shown potential long-lasting impact of cardiomyocytes when comparing to the basal 1 day controls. Due to the original study design we don’t have the optimal controls to compare to every time point, but since there are relevant biological changes occurring when compared to the basal (1 day) ground controls we hypothesis that there will be consistent long-lasting impact on cardiomyocytes isolated from mice irradiated with protons and 56Fe ion irradiation. For the key gene analysis, we pooled all irradiated time-point together to determine the common key driving genes for each type of radiation.”

 p. 13 lines 486-487: “Pathway analysis and subsequent predictions in each tissue were done using the statistically significant genes with a fold-change ≥1.2 (or ≤-1.2)”. Even if the changes were statistically significant, what is the biological relevance of such small gene expression changes?

 Fold-change values are arbitrary values that are up to the discretion of the investigator. A 1.2 fold-change value is very valid to use. As we (and other investigators) have shown, small fold-changes are important to consider. From our past experience it isn’t the individual genes that are important, but the overall group of genes that changes that will impact entire pathways that cause the most biological changes in a system. This is a standard viewpoint for a systems biology type of analysis that we have implemented in this analysis. So, although a smaller fold-change might not seem important at the individual gene level, a group of genes with smaller fold-changes can cause large dysregulation with relevant pathways affecting the system a person is studying. Using larger fold-change values will impact the predictions for relevant pathways in the downstream analysis that will be relevant to the data. We have added the following text to address this point after the above statement in the manuscript:

 “Using a 1.2 fold-change cutoff will allow us to determine the impact of how major pathways are being regulated that will be not be apparent with larger fold-change cutoffs. It has been shown in the literature that larger fold-change values used for Omics data will significantly reduce useful biological data that becomes apparent when using lower fold-change cutoffs [77,78].”

 Results:

p. 3 line 121 GLDS-52 => here, short information on the controls (inflight 1 x g centrifuge, ground controls …) is missing.

 We added the following sentence to address this comment:

 “The HUVEC controls for the GLDS-52 datasets were cultured for the same length and methodology as the samples flown on the ISS, but were done on Earth [44].”

  p. 3 line 126-127 “for the proton study (GLDS-117) there were 2540 significantly regulated genes” => it is unclear for which time point this applies.

 Since we used ANOVA analysis for the ground studies (both protons and 56Fe) all time points have to be used to produce statistically significant genes. When determining statistically significant genes from an experiment with multiple related data points it is standard procedure to use ANOVA to determine the statistically significant genes across all datapoints. We have modified the sentence the following way to indicate this:

 “We used either one-way ANOVA analysis for the simulated space radiation studies (all time points were used to determine the significant genes) or t-test analysis with a p-value ≤ 0.05 for the spaceflight study.”

 p. 3 line 126-127         “for the 56Fe study (GLDS-109) there were 5554 significantly regulated genes” => it is unclear for which time point this applies.

 See reply to the previous comment.

 p. 4 line 171    “estimated dose 1.544 mGy” => To my opinion, there are too many digits for an estimation only.

 1.544 mGy is directly from Thomas Berger, but looking at similar published results, the typical uncertainty is 8-10%, so it is reasonable to change to 1.5 mGy.  The reference for data taken on Columbus in that period is: T. Berger et al., J. Space Weather Space Clim. 7 A8 (2017). We have changed the text to include from 1.544 mGy to 1.5 mGy.

 Supplemental Figure:

I think the figure should be included in the manuscript as it does not take much space and is discussed in detail and it is inconvenient for the reader to have to download it separately.

 We have included the supplemental figure as a figure for the manuscript.

 Materials and Methods:

 This part lacks some important information:

 p. 12 line 471 “cardiomyocytes were isolated” => Did the researchers take the cardiomyocytes in culture? If, for which time period? Or was it a tissue dissection in order to remove other cell types?

 We have clarified this point by adding the following statement to this method section:

 “All cardiomyocytes that were isolated were from tissue dissections in order to the remove from other cell types.”

 p. 12 line 473-475 It is unclear which controls (1 x g centrifuge on ISS ?, ground controls?) were used in this spaceflight experiment.

 For these experiments, the 1xg controls are the controls. There were no other controls available from the ISS from the original study. We have added the following statement in the methods to address this comment:

 “The ground controls used for this study are 1 × g controls that were grown on Earth with identical conditions compared to the cells on the ISS.”

 Minor comments:

p. 1 line 16                  “male C57BL/6” => “male C57BL/6 mice”

 We made this change.

 p. 1 line 16      “90cGy” => cGy is not a SI unit and should not be used. Please use Gy or mGy (check the whole manuscript).

 We converted all values with cGy to mGy throughout the manscript.

 p. 1 line 19      “FYN” => please explain all abbreviations at their first appearance in the abstract and in the text.

 FYN is the gene name and is standard to use the gene name when discussing it.

 p. 1 line 23      “Protons” => “protons”

 We made this change.

 p. 1 line 34                  “LEO (Low Earth Orbit)” => “Low Earth Orbit (LEO)”

 We made this change.

  p. 1 line 38                  “earth” => “Earth”

 We made this change.

 p. 2 line 53                  “earth” => “Earth”

 We made this change.

 p. 3 line 104                “earth” => “Earth”

 We made this change.

 p. 3 line 107                “earth” => “Earth”

 We made this change.

 p. 5 line 184:   “2.2. .Dose Received by HUVECs Flown in Space” => “2.2. Dose Received by HUVECs Flown in Space”

 We made this change.

 p. 5 line 206    “reactive oxygen species (ROS)” => “ROS”

 We made this change.

 p. 5 line 208    “on the cardiovascular systems” => “on the cardiovascular system”

 We made this change.

 p. 5 line 213    “ROS has also been shown” => “ROS have also been shown”

 We made this change.

 p. 5 line 215    “ROS has been shown” => “ROS have been shown”

 We made this change.

 p. 5 line 217:   “that ROS is having long lasting effects” => “that ROS are having long lasting effects”

 We made this change.

 p. 5 line 218:   “DNA repair Hallmarks” => “DNA repair hallmarks”

 We made this change.

 p. 7 line 245: “2.4. The Biological Pathways Elicited by Protons have stronger overlap with Spaceflight Samples than 56Fe” => “2.4. The Biological Pathways Elicited by Protons have stronger overlap with Spaceflight Samples than those elicited by 56Fe ions”

 We made this change.

 p. 7 line 246 “one type of space irradiation” => “one type of space radiation”

 We made this change.

 p. 8 line 297 “56Fe irradiation” => “56Fe ion irradiation” (check the whole manuscript)

 The common usage is not to have “ion”, but we have made these changes throughout the manuscript for clarity for the non-radiation audience.

 p. 8 line 305 “time points” => time-points (spelling varies within the manuscript, please check)

 We changed time points to time-points throughout the manuscript.

 p. 8 line 317 “Interestingly it known” => “Interestingly it is known”

 We made this change.

 p. 9 line 325 “leads to acitvation” => “leads to activation”

 We made this change.

 p. 9 line 336 “the two irradiated conditions” => “the two irradiation conditions”

 We made this change.

 p. 9 line 336 “the colors in the top half indicates” => “the colors in the top half indicate”

 We made this change.

 p. 12 line 466 “1GeV” => “1 GeV”

 We made this change.

 p. 14 line 539 “Hze” => “HZE”

 We made this change.

 p. 14 line 542 “mars” => “Mars”

 We made this change.

 p. 15 line 568 “Nasa genelab” => “NASA GeneLab”

 We made this change.

 p. 15 line 573 “(vegf)” => “(VEGF)”

 We made this change.

 p. 15 line 588 “si” => “Si”

 We made this change.

 p. 15 line 590 “let” => “LET”

 We made this change.

 p. 15 line 590 “fe” => “Fe”

 We made this change.

 p. 15 line 593 “apolipoprotein e” => “apolipoprotein E”

 We made this change.

 p. 15 line 610 “fyn” => “FYN”

 We made this change.

 p. 16 line 619 “nrf2” => “Nrf2”

 We made this change.

 p. 16 line 632 “acetyl-coa” => “acetyl-coA”

 We made this change.

 p. 16 line 638 “microrna” => “microRNA”

 We made this change.

 p. 16 line 638 “tgf-beta1” => “TGF-beta1”

 We made this change.

 p. 16 line 656 “t-cell” => “T-cell”

 We made this change.

 p. 16 line 658 “fyn” => “FYN”

 We made this change.

 p. 16 line 660 “src-family kinases, lck and fyn, in t-cell” => “SRC-family kinases, LCK and FYN, in T-cell”

 We made this change.

 p. 17 line 678 “Elevating cxcr7 improves angiogenic function of epcs via akt/gsk-3beta/fyn-mediated nrf2 activation” => “Elevating CXCR7 improves angiogenic function of EPCs via Akt/GSK-3beta/FYN-mediated Nrf2 activation”

 We made this change.

 p. 17 line 681 “nrf2 suppression via modulation of akt/gsk3beta/fyn kinase” => “Nrf2 suppression via modulation of Akt/GSK-3beta/FYN kinase”

 We made this change.

 p. 17 line 689 “Micrornas” => “MicroRNAs”

 We made this change.

 p. 17 line 696 “nrf2” => “Nrf2”

 Reference list: check journal abbreviations / names

We thoroughly went through references and checked the journal abbreviations/names.

Reviewer 3 Report

The manuscript entitled “Space Radiation induces long term impact on the Cardiovascular System by the activation of FYN through Reactive Oxygen Species” by Beheshti et al. claims irradiation causes some alteration to cardiovascular system via pathway related to reactive oxygen species, and FYN is the gene involved in it. However, there are many concerns that should be addressed appropriately before it is published and regarded as universal scientific knowledge.

 The most important concern, which is implied by the authors themselves, is that the gene expression data lacks proper time-matched control. Without this, any detected alterations cannot be discriminated from the effect of time. Time-matched control should be included, especially when we consider the “long term impact” in response to any conditions.

According to Fig. 1B, the values of the principal component 1 (PC1) from 0Gy controls between 0 days (black squares: ~-10 and -40) and 3 days (gray square: ~-150) seem to be obviously different from each other, which is also implied in Fig. 1E.

 Line 31. “Long-duration space travel has been associated with a number of human health risk factors, such as space radiation, microgravity, isolation, and hypoxia [1-3].”

—The fact that the long-duration space travel has been associated with hypoxia does not seem to be found in the references 1-3. Please give another reference.

 Line 93. “The two main components that make up the cardiac tissue are cardiomyocytes and endothelial cells”

—This statement would be oversimplifying the structure of the heart according to the fact that cardiac fibroblasts are the most abundant cell in the mammalian heart [J Mol Cell Cardiol 70, 2-8, doi:10.1016/j.yjmcc.2013.11.003 (2014)]. Besides, not only are they abundant, cardiac fibroblasts are involved with cardiomyocyte functions [Open Biol 5, 150038, doi:10.1098/rsob.150038 (2015)].

 Line 95. “the HUVEC response to spaceflight should be highly correlated with cardiomyocytes.”

—The endothelial cells predominate in the cardiac tissue would be cardiac microvascular endothelial cells. According to Chi et al., gene expression pattern among endothelial cell lines are diverse [Proc Natl Acad Sci U S A 100, 10623-10628, doi:10.1073/pnas.1434429100 (2003)]. It is even likely that the gene expression pattern of HUVEC is distinct from that of coronary artery endothelial cells (Fig. 1 of the reference).

 Line 159. “It is interesting to note the samples that produced the greatest separation were the cardiomyocytes isolated at the longest time points (either 26 or 28 days) after irradiation.”

—It is difficult to confirm this statement from Figs. 1A and 1B, which do not show the maximum difference of PC1 value between 26/28 days and control.

 Line 180. “There also is a distinct group of genes that being regulated oppositely for HUVECs flown in space compared to the ground controls.”

—It is rather usual that there are both upregulated and downregulated genes in response to a certain condition when we run this kind of gene analysis. Is there any further implication here?

 Line 189. “The measured dose rate was almost constant throughout the experiment”

—It would be better if the authors could show the original time-course of the measurement.

 Line 204. “most time points”

—These “time points” presumably correspond to the “11 wedges for each condition” described in the legend of Fig. 2. It would be appropriate to explicitly indicate the 11 conditions, and evaluate the effect of time after irradiation, since the focus of this research is on the “long term impact” of the space radiation.

 Line 206. “reactive oxygen species (ROS) pathways are surprisingly downregulated”

—Is the ROS production increased or decreased when the ROS pathways are downregulated?

 Line 209. “The other pathways that were downregulated were also connected to the ROS pathway. For example, Fatty Acid Metabolism, which is one of the primary sources of energy for cardiac muscle, has direct link to ROS formation and regulation through the mitochondria (Figure 2) [40].”

—These sentences imply that the ROS pathway was connected to the fatty acid metabolism pathway, and that the fact is shown in Fig. 2. There are two concerns. First, we cannot find direct link between the ROS pathway and the fatty acid metabolism pathway in Fig. 2. Second, the “direct link” between the fatty acid metabolism and the ROS formation discussed in [40] is about mitochondrial β-oxidation, but we are not sure whether the “fatty acid metabolism” shown in Fig. 2 is (or contains at least) mitochondrial β-oxidation.

 Line 211. “Knocking out MYC has been shown to cause low levels of glycolysis and oxidative phosphorylation”

—What is the relationship between “knocking out MYC” and the main claim of this research?

 Line 214. “The downregulation of the ROS seems to also lead overall downregulation of cell cycle pathways (Supplemental Figure 1).”

—Where can we find the ROS pathway in Supplemental Fig. 1?

 Line 217. “we show that ROS is having long lasting effects directly linked to the space environment on the cardiovascular systems.”

—From which data can we see this “long lasting effect”?

 Line 228. “Note, most of them are up-regulated.” (Fig. 2C)

—What is the relationship between this fact and the main claim of this research?

 Line 230. “Note, most of them are down-regulated.” (Fig. 2D)

—Half of the wedges in the circles are red, which suggests that they are upregulated. Besides, there seems to be wedges in orange, which is not in the color scale bar. What do they mean?

 Line 247. “Figure 2C shows that the majority of pathways specifically in common with protons and spaceflight were upregulated while Figure 2D shows the 56Fe component in common with spaceflight was downregulated.”

—It is hard to follow this statement. Frequency of red and blue seems almost the same both in Figs. 2C and 2D. If not, please show us the exact number and statistical evidence.

 Line 254. “In general, the overall pathways and functions overlap with the GSEA findings.”

—It is hard to follow this statement. Please indicate the names of overlapped pathways and functions.

 Line 269. “specific components are showing pathways and functions related to ROS and other key factors impacting the cardiovascular system”

—Where can we find this fact?

 Line 274. “we show unequivocally that all cardiomyocyte samples irradiated with protons have biofunctions, upstream regulators and canonical pathways which cluster closely to the spaceflight samples (Figures 3D – 3F).”

—At first, Z-scores of spaceflight sample (yellow) are almost or nearly zero in Figs. 3A and 3B, which suggests that spaceflight did not have significant effect on Biofunctions and Upstream Regulators. This inference is supported by the fact that the PC1 and PC2 values of the spaceflight sample in Figs. 3D and 3E are almost zero. On the other hand, all the purple signs (Proton irradiated group) are located near (0,0), which is overlapped with the position of the spaceflight sample, unequivocally throughout Figs. 3D-F. Based on these facts, it might be rather the spaceflight and proton-irradiated samples did not have any significant biofunctions.

 Line 278. “7 days and 28 days seem to behave as outliers when compared to the other datasets.”

—We have to be careful to exclude two samples as outliers out of five samples. If we exclude these samples as outliers, then it will be difficult to ensure the validity of all the other results using these time points (especially 28 days), which could jeopardize the claim of “long term impact.”

 Line 314. “FYN being the overall central driver/hub for the cardiovascular response to space radiation (Figure 4A).”

—From which evidence is this statement confirmed in Fig. 4? With regard to the key genes in HUVEC, connections to FYN seem to be mostly “Effect not predicted” or “Indirect Relationship” in Fig. 6.

 Line 317. “Interestingly it known that LCK, LYN, and FYN are part of the SRC-family kinases (SFKs) [52-54] and have been associated with various aspects of cardiovascular disease [54-58].”

—According to the reference 54, Lyn is indeed associated with cardiovascular disease, as phosphorylation of Lyn leads to activation of Na+/K+ ATPase and subsequent foam cell formation that causes atherosclerosis. However, it is about macrophage and not about cardiac cells used in the authors’ study.

Author Response

December 13th, 2018

 Re: addressing reviewers comments

 Dear Reviewer 3,

 We thank you for providing thoughtful comments for our manuscript. I have provided the revised manuscript with track changes so the editors and reviewers can easily view the requested changes. We have also provided revisions for the following figures based on the comments provided by the reviewers: Figure 01, Figure 02, Figure 05, Figure 07, and the Graphical Abstract. In addition, we have now included the Supplemental Figure as a figure in the manuscript based on Reviewer #2’s comments. This figure is now labeled as Figure 03 and all figure labels have been adjusted accordingly throughout the manuscript.

 Below we have provided a detailed response to your comments and concerns. We have provided both the reviewer’s comments and our responses to each comment in red text. We believe we have adequately responded to the reviewers comments and look forward to hearing back from you.

 Afshin Beheshti, PhD

 Reviewer #3

 Comments and Suggestions for Authors

The manuscript entitled “Space Radiation induces long term impact on the Cardiovascular System by the activation of FYN through Reactive Oxygen Species” by Beheshti et al. claims irradiation causes some alteration to cardiovascular system via pathway related to reactive oxygen species, and FYN is the gene involved in it. However, there are many concerns that should be addressed appropriately before it is published and regarded as universal scientific knowledge.

 The most important concern, which is implied by the authors themselves, is that the gene expression data lacks proper time-matched control. Without this, any detected alterations cannot be discriminated from the effect of time. Time-matched control should be included, especially when we consider the “long term impact” in response to any conditions.

 We agree with the reviewer that better controls should have been implemented in the original study by the original investigators. Unfortunately, the poor setup of the original experiment is out of our control. The point of this study is to show how GeneLab datasets can be used to generate novel and interesting hypothesis to guide future space biology experiments. As stated in the manuscript based on the controls provided by the dataset, we are still able to find relevant biology that agreed with what is known in the literature that can be applied to space radiation effects for cardiovascular disease. We have indicated this in the manuscript (page 4 line 171).. We also have added the following text in that section to further explain this point (page 3 line 125):

 “Although this is not the most optimal setup for these types of experiments, we will show that former datasets from GeneLab without the desired experimental setup can still produce meaningful results to guide space biology research.”

 In addition, we believe that comparing to the original basal controls (1 day control group) will at least allow us to determine how the biology changes when compared to the original start of the experiment. We have rephrased sentences addressing this part to indicate this (page 9 line 412):

 “Our data has shown potential long-lasting impact of cardiomyocytes when comparing to the basal 1 day controls. Due to the original study design we don’t have the optimal controls to compare to every time point, but since there are relevant biological changes occurring when compared to the basal (1 day) ground controls we hypothesis that there will be consistent long-lasting impact on cardiomyocytes isolated from mice irradiated with protons and 56Fe ion irradiation. For the key gene analysis, we pooled all irradiated time-point together to determine the common key driving genes for each type of radiation.”

 According to Fig. 1B, the values of the principal component 1 (PC1) from 0Gy controls between 0 days (black squares: ~-10 and -40) and 3 days (gray square: ~-150) seem to be obviously different from each other, which is also implied in Fig. 1E.

 We observed this also with our analysis. Although for the 56Fe 0 Gy controls this seems to be the case, the proton 0Gy controls all seem to cluster closer together and behave overall the same (Figs. 1A and 1D). Since we want to analyze the datasets consistently, we used these “basal” controls with the same methodology throughout our analysis. As stated above unfortunately the original study design is out of our hand, but we believe that the results from our analysis are still useful to form conclusions.

 Line 31. “Long-duration space travel has been associated with a number of human health risk factors, such as space radiation, microgravity, isolation, and hypoxia [1-3].”

—The fact that the long-duration space travel has been associated with hypoxia does not seem to be found in the references 1-3. Please give another reference.

 We have added a reference for hypoxia and space travel.

 Line 93. “The two main components that make up the cardiac tissue are cardiomyocytes and endothelial cells”

—This statement would be oversimplifying the structure of the heart according to the fact that cardiac fibroblasts are the most abundant cell in the mammalian heart [J Mol Cell Cardiol 70, 2-8, doi:10.1016/j.yjmcc.2013.11.003 (2014)]. Besides, not only are they abundant, cardiac fibroblasts are involved with cardiomyocyte functions [Open Biol 5, 150038, doi:10.1098/rsob.150038 (2015)].

 Due to the availability of data associated with the cardiovascular tissue we have only focused on these two components. We have modified the text for this part to reflect the reviewers comments. The following is the modified text:

 “Two components that make up the cardiac tissue are cardiomyocytes and endothelial cells and both cell types are heavily involved with cardiac remodeling and regeneration [32]. Although there are more components to the cardiac tissue, from the datasets available from GeneLab related to the cardiovascular, we will only focus on these components.”

 Line 95. “the HUVEC response to spaceflight should be highly correlated with cardiomyocytes.”

—The endothelial cells predominate in the cardiac tissue would be cardiac microvascular endothelial cells. According to Chi et al., gene expression pattern among endothelial cell lines are diverse [Proc Natl Acad Sci U S A 100, 10623-10628, doi:10.1073/pnas.1434429100 (2003)]. It is even likely that the gene expression pattern of HUVEC is distinct from that of coronary artery endothelial cells (Fig. 1 of the reference).

 To address this comment, we have removed the sentence.

 Line 159. “It is interesting to note the samples that produced the greatest separation were the cardiomyocytes isolated at the longest time points (either 26 or 28 days) after irradiation.”

—It is difficult to confirm this statement from Figs. 1A and 1B, which do not show the maximum difference of PC1 value between 26/28 days and control.

 For Fig 1A we see the maximum difference occurring in the PC2 axis between 26 days and controls. For Fig 1B we see that the 28 days in general is separated farther in the negative PC1 axis compared to all other samples. The separation between these time points is further evident in the clustering occurring with the significantly regulated genes (Figs. 1D and 1E). We have added the following text to further explain this (page 4 line 206):

 “This is apparent in the PC2 axis for protons (Figure 1A) and for 56Fe ions these points are clustering further apart in the negative PC1 axis from all other samples.”

 Line 180. “There also is a distinct group of genes that being regulated oppositely for HUVECs flown in space compared to the ground controls.”

—It is rather usual that there are both upregulated and downregulated genes in response to a certain condition when we run this kind of gene analysis. Is there any further implication here?

 There are no further implications. This statement and part were just to set the groundwork to show that there are distinct global changes happening between the experimental conditions and the controls. If there were not any global changes occurring then no further analysis would not have happened.

 Line 189. “The measured dose rate was almost constant throughout the experiment”

—It would be better if the authors could show the original time-course of the measurement.

 To address this comment, we have provided the average dose rate amount with the standard deviation. The standard deviation is strikingly a small value which will indicate the relative constant measured dose-rate. We have added the following to address this point (page 5 line 239):

 “The measured dose rate was almost constant throughout the experiment with an average of 220.43 µGy/day (in water) with a standard deviation of 1.90 µGy/day (in water) measured over the period of the experiment (Data provided kindly by T. Berger from the DLR and additional details can be found in the original publication [45]).”

 Line 204. “most time points”

—These “time points” presumably correspond to the “11 wedges for each condition” described in the legend of Fig. 2. It would be appropriate to explicitly indicate the 11 conditions, and evaluate the effect of time after irradiation, since the focus of this research is on the “long term impact” of the space radiation.

The time-points are the different time-points after irradiation as explained in the beginning of the result section. The figure legend for figure 2 distinctly shows which wedge of each node is associated with the specific time-point and sample. We have modified and added the following to address this comment (page 5 line 256):

 “We first observed that the majority of pathways for most time-points after radiation for the cardiomyocytes that were commonly regulated with the HUVEC spaceflight samples were downregulated (with exception of one time point at 1 day for samples irradiated with protons for some nodes) (Figure 2A). The time-points and samples are associated with each wedge in the node as displayed in the figure legend for Figure 2.”

 Line 206. “reactive oxygen species (ROS) pathways are surprisingly downregulated”

—Is the ROS production increased or decreased when the ROS pathways are downregulated?

 The ROS production will be decreased with the pathways are downregulated. We have added the following to clarify this point (page 5 line 261):

 “Specifically, ROS pathways are surprisingly downregulated (indicating decrease in ROS production) for the majority of the conditions,…”

 Line 209. “The other pathways that were downregulated were also connected to the ROS pathway. For example, Fatty Acid Metabolism, which is one of the primary sources of energy for cardiac muscle, has direct link to ROS formation and regulation through the mitochondria (Figure 2) [40].”

—These sentences imply that the ROS pathway was connected to the fatty acid metabolism pathway, and that the fact is shown in Fig. 2. There are two concerns. First, we cannot find direct link between the ROS pathway and the fatty acid metabolism pathway in Fig. 2. Second, the “direct link” between the fatty acid metabolism and the ROS formation discussed in [40] is about mitochondrial β-oxidation, but we are not sure whether the “fatty acid metabolism” shown in Fig. 2 is (or contains at least) mitochondrial β-oxidation.

  It has been shown from past research that potentially the majority of the ROS production for cardiac tissue is produced by the mitochondria. So although it is correct to say that we haven’t proven a direct link there definitely is a correlation between what we have observed regarding this comment. We have added the following comment to further explain the relevance (page 5 line 267):

 “It is interesting to note that it has been reported that the ROS production from the mitochondria is an important mechanism of how ROS impacts cardiac tissue [49].”

 Line 211. “Knocking out MYC has been shown to cause low levels of glycolysis and oxidative phosphorylation”

—What is the relationship between “knocking out MYC” and the main claim of this research?

 This is related to the research claims since oxidative phosphorylation is known to produce ROS such as superoxide and hydrogen peroxide. The suppression through MYC can potentially also provide another avenue of downregulated ROS pathway. We have added the following to address this comment (page 6 line 285):

 “It is known that oxidative phosphorylation will produce ROS [50] and with the downregulation of MYC this can potentially reduce oxidative phosphorylation levels which in turn will reduce ROS.”

 Line 214. “The downregulation of the ROS seems to also lead overall downregulation of cell cycle pathways (Supplemental Figure 1).”

—Where can we find the ROS pathway in Supplemental Fig. 1?

 This statement we made is more implying that we see downregulation of cell cycle. This might be caused by the overall downregulation of ROS. The reviewer is correct there is no direct relationship. We have changed the above statement to the following to reflect this (page 6 line 288):

 “We also observe an overall downregulation of cell cycle pathways (Figure 3). We believe that this might be potentially due to the downregulation of ROS signaling, since it has been previously shown that ROS can heavily influence cell cycle [52].”

 Line 217. “we show that ROS is having long lasting effects directly linked to the space environment on the cardiovascular systems.”

—From which data can we see this “long lasting effect”?

 We observe in figure 2 that regardless of the time points ROS is still downregulated long after the acute radiation given. This indicates to us that it is reasonable to say that from the basal level ROS levels are being consistently downregulated up to the time points available from the datasets.

 Line 228. “Note, most of them are up-regulated.” (Fig. 2C)

—What is the relationship between this fact and the main claim of this research?

 We didn’t discuss this here, since this is the figure legend. We have discussed this relationship fully in results section which is related to this figure legend and comment.

 Line 230. “Note, most of them are down-regulated.” (Fig. 2D)

—Half of the wedges in the circles are red, which suggests that they are upregulated. Besides, there seems to be wedges in orange, which is not in the color scale bar. What do they mean?

 This is specifically referring to panel 2D which is stating the majority of the pathways that are downregulated between 56Fe and spaceflight samples. We have modified the above sentence the following way to clarify (page 6 line 307):

 “Note, most of them are down-regulated for 56Fe and Spaceflight samples.”

 In addition, there are no orange wedges in the figure. There are different shades of red indicating the amount of regulation. At the end of the figure legend we have stated this (page 7 line 332):

 “The shade of the color indicates degree of regulation.”

 Line 247. “Figure 2C shows that the majority of pathways specifically in common with protons and spaceflight were upregulated while Figure 2D shows the 56Fe component in common with spaceflight was downregulated.”

—It is hard to follow this statement. Frequency of red and blue seems almost the same both in Figs. 2C and 2D. If not, please show us the exact number and statistical evidence.

 Figure 2C is only relating protons to spaceflight and Figure 2D is only comparing 56Fe with spaceflight. This why the heading is only referring to the pathways that the specific ground experiment is in common with spaceflight associated pathways. We have added yellow highlights around the wedges that should be the focus for those panels. We also added the text to the figure caption (page 6 line 308):

 “For figures (C) and (D) the common wedges are highlighted in yellow outline.”

 The key difference between those two panels is which direction the pathway is associated with spaceflight and not what the majority of all pathways are doing.

 Line 254. “In general, the overall pathways and functions overlap with the GSEA findings.”

—It is hard to follow this statement. Please indicate the names of overlapped pathways and functions.

 We add the following text after this sentence to address the reviewers comments (page 7, line 343):

 “For example, we observe in the upstream regulator predictions that MYC is downregulated for the majority for the pathways. We also observe that the following upstream regulators predicted to the activate or inhibited in the same directions as the GSEA predictions: p53, HIF1A, IL2, and IL6. The mTOR signaling Canonical Pathway also overlaps with the GSEA predictions.”

 Line 269. “specific components are showing pathways and functions related to ROS and other key factors impacting the cardiovascular system”

—Where can we find this fact?

 This was a lead into sentence to the next paragraph where we want to focus on how the overall functions from the ground studies group together with spaceflight samples. To avoid taking focus away from the point of this paragraph we removed the beginning part of this sentence to only focus on what relevant for this paragraph. The following is how the sentence now reads (page 8, line 371):

 “We were interested in determining how these overall functions from the ground experiments will group with the spaceflight samples.”

 Line 274. “we show unequivocally that all cardiomyocyte samples irradiated with protons have biofunctions, upstream regulators and canonical pathways which cluster closely to the spaceflight samples (Figures 3D – 3F).”

—At first, Z-scores of spaceflight sample (yellow) are almost or nearly zero in Figs. 3A and 3B, which suggests that spaceflight did not have significant effect on Biofunctions and Upstream Regulators. This inference is supported by the fact that the PC1 and PC2 values of the spaceflight sample in Figs. 3D and 3E are almost zero. On the other hand, all the purple signs (Proton irradiated group) are located near (0,0), which is overlapped with the position of the spaceflight sample, unequivocally throughout Figs. 3D-F. Based on these facts, it might be rather the spaceflight and proton-irradiated samples did not have any significant biofunctions.

 For PCA plots a value of 0 on the axis does not equate to no significant value for the specific function or data being plotted. It refers to the amount of variance. For example, for the proton irradiated samples, the biofunctions and upstream regulator predictions have a plenty of significantly regulated factors (indicated by the z-scores, see figures 4A – 4C). But since the functions are showing less variance between the sample and between the specific functions the samples will cluster closer to 0 and also all time-points will cluster closer together. For 56Fe samples it is evident from the heatmaps that there is a lot more variance occurring between different samples and also across the functions. This is why we observe larger PC values associated with the datapoints associated with 56Fe.

 You also have to keep in mind that I am showing only the functions that are associated with the top 4 prevalence of change in the heatmaps (i.e. prevalence of change ≥ 7). For the PCA plots I used all the functions with a z-score. So, there will be other functions not present in the heatmaps that were used for the PCA plot calculations. I have added the following to the figure captions for further clarification (page 8, line 366):

 “For all heatmaps we are only showing the prevalence of change for the functions associated with the top 4 (i.e. a prevalence of change score ≥ 7).”

 And (page 8 line 369)

 “For the PCA plots we used all functions for each datapoint regardless of the prevalence of change score.”

 Line 278. “7 days and 28 days seem to behave as outliers when compared to the other datasets.”

—We have to be careful to exclude two samples as outliers out of five samples. If we exclude these samples as outliers, then it will be difficult to ensure the validity of all the other results using these time points (especially 28 days), which could jeopardize the claim of “long term impact.”

 We agree with the reviewer with this statement. Which is why in the next section we did analysis showing both with these time-points and without. That way we can show the readers the key driving genes that will be present with and without these potential outliers.

 Line 314. “FYN being the overall central driver/hub for the cardiovascular response to space radiation (Figure 4A).”

—From which evidence is this statement confirmed in Fig. 4? With regard to the key genes in HUVEC, connections to FYN seem to be mostly “Effect not predicted” or “Indirect Relationship” in Fig. 6.

 This evidence is provided by the interactions between all the actual key genes in Figure 5A. FYN in has the most connections between all the key genes in this network. How the network is made is through what is known in the literature. The term “Effect not predicted” means that it is known in the literature that FYN has a relationship with the connection gene, but it is not known if this gene will interact with the other. The term “Indirect Relationship” is defined as the interaction between the two genes is known to have an impact on each other, but the mechanism can be through indirect avenues. But the influence between the two genes are still significant according to the literature to warrant a connection in the network. These terms should not be interpreted as the connection between the two genes are not meaningful, but on the influence on how the genes will interact in the network. The important point is not to focus on these terms, but to focus on how many connections are made between the genes. The gene with the most connections can be thought of as the central hub. Once can think if this central hub is removed from the network, now the system (or as an example the circuitry) will not function optimally anymore due to the central being removed. We have shown in previous publications that by knocking out similar central hubs in the system being studied, does indeed functionally impair the expected outcome.

 Line 317. “Interestingly it known that LCK, LYN, and FYN are part of the SRC-family kinases (SFKs) [52-54] and have been associated with various aspects of cardiovascular disease [54-58].”

—According to the reference 54, Lyn is indeed associated with cardiovascular disease, as phosphorylation of Lyn leads to activation of Na+/K+ ATPase and subsequent foam cell formation that causes atherosclerosis. However, it is about macrophage and not about cardiac cells used in the authors’ study.

 We have added an additional reference that shows the direct interaction between LYN and FYN on specifically the endothelial cells (more specifically HUVECs).

Round  2

Reviewer 2 Report

Most major and all minor comments were fully addressed in the author’s response. Unfortunately, the file with the track changes was not available, so I couldn’t verify all the changes in the manuscript. Strong concerns remain on the conclusions from the 1.5 mGy – 900 mGy proton exposure comparison and for the long-term effects as the adequate controls are missing, and especially as the long-term effects are stated in the title of the paper. p. 8 line 284-285: “These results indicate that the overall functional impact of space radiation on the ISS on samples is due to protons rather than HZE ions.” p. 10 line 381-382: “1) that, consistent with the dose data, the majority of the radiation experienced on the ISS is from protons” p. 12 Lines 450-452: “By creatively combing several datasets we were able to isolate for the first time the specific space radiation component that astronauts are being exposed to at LEO on the ISS.” => Regarding the difference in total dose (1.5 mGy accumulated on the ISS within 7 days, and 900 mGy protons in the accelerator experiment), I don’t think that such a conclusion is possible. Here, a comparison of the gene expression profile after incubation on the ISS and exposure on ground to 1.5 mGy protons at an accelerator would be necessary. The influence of other spaceflight factors including microgravity and the experimental conditions on the ISS might play a much higher role than the tiny radiation dose. Also, different cell types are compared by this approach (cardiomyocytes and endothelial cells). These many influencing factors do not allow the conclusions that are cited above. Especially as the results are suggested to be considered in risk models, high care has to be taken that the data support the conclusions. We agree with the reviewer that further experiments will have to be done fully validate these findings. We believe that the power of this type of analysis with publicly available datasets from GeneLab, allows the public to potentially determine the best future experiments to perform. As stated in the manuscript we believe, since the microgravity component is not present with the mice ground experiments it provides a nice way to only dissect the radiation component when comparing to spaceflight studies. We believe that these results can provide some evidence of the specific space radiation component experienced at LEO. That said, we have rephrased our statements regarding this issue to provide the “potential” that protons are the majority of the type of radiation experiences at LEO. Here are the rephrased sentences: “1) that, consistent with the dose data, the majority of the radiation experienced on the ISS is potentially from protons” “By creatively combing several datasets we have potential indication for the first time of the specific space radiation component that astronauts are being exposed to at LEO on the ISS. It has been previously thought that protons comprise the majority of ions that astronauts are exposed to in LEO [15], but biological data have been lacking. Of course, further experiments have to be performed to determine whether these findings are true, but the power of this type of analysis with GeneLab datasets allows for future guidance for space biology related experiments. Abstract: “To our knowledge, this is the first biological confirmation that the majority of ions on the ISS are protons, clearly illustrating the power of Omics analysis.” Again, these conclusions are not supported by the data and are misleading. The conclusion can be, that similar transcriptomic signatures are observed after 900 mGy proton exposure on ground and by 7 d cultivation of cells on the ISS, meaning exposure to 1.5 mGy protons, microgravity and other experimental conditions on the ISS. It can be suggested to examine the contribution of the 1.5 mGy protons to the gene expression changes observed on ISS by accelerator experiments on ground. p. 2 line 77-80 “Specifically, it was shown that the mortality rate due to cardiovascular disease for astronauts who flew to the Moon was approximately 20% higher than that in the normal US population and that exposure to space radiation causes long lasting vascular endothelial dysfunction [30].” => This article has provoked a lot of critics and it was shown that the conclusions are not supported by the results presented in the paper. It should not be cited in such an uncritical way. Cucinotta FA, Hamada N, Little MP. No evidence for an increase in circulatory disease mortality in astronauts following space radiation exposures. Life Sci Space Res (Amst). 2016 Aug;10:53-6. doi: 10.1016/j.lssr.2016.08.002. We thank the reviewer to bringing this to our attention and we have removed this reference and sentence from our manuscript. But there is still the sentence: “More recently, data from astronauts involved with NASA’s Apollo program has indicated long lasting impacts on the cardiovascular system [20,31]. The Takahashi paper cites the Delp Apollo paper (Sci. Rep.2016, 6, 29901) and has no own data on cardiovascular effects in Apollo astronauts. The Hughson paper is a review paper that also contains no better Apollo astronaut data but it discusses the cardiovascular risk by low dose radiation exposure in the context of spaceflight, meaning the cardiovascular adaptation to microgravity. p. 8 lines 296-297: “Since our data has shown consistent long-lasting impact on cardiomyocytes isolated from mice irradiated with protons and 56Fe irradiation” => As the controls for the late time-points are missing, no conclusions about long-lasting effects are possible. This is especially true as the 7 and 28 days results are outliers. Although the original study design isn’t optimal, we believe that comparing to the original basal controls (1 day control group) will at least allow us to determine how the biology changes when compared to the original start of the experiment. We have rephrased this part to indicate this: “Our data has shown potential long-lasting impact of cardiomyocytes when comparing to the basal 1 day controls. Due to the original study design we don’t have the optimal controls to compare to every time point, but since there are relevant biological changes occurring when compared to the basal (1 day) ground controls we hypothesis that there will be consistent long-lasting impact on cardiomyocytes isolated from mice irradiated with protons and 56Fe ion irradiation. For the key gene analysis, we pooled all irradiated time-point together to determine the common key driving genes for each type of radiation.” I strongly recommend being more careful with the conclusions for the late time-points, especially not to highlight them in the title, as the correct control is missing. A literature / database search on gene expression in murine cardiomyocytes (in vivo) over a time period of month might help to reduce the uncertainty. p. 12 line 471 “cardiomyocytes were isolated” => Did the researchers take the cardiomyocytes in culture? If, for which time period? Or was it a tissue dissection in order to remove other cell types? We have clarified this point by adding the following statement to this method section: “All cardiomyocytes that were isolated were from tissue dissections in order to the remove from other cell types.”  “Cardiomyocytes were isolated by microdissection from tissue dissections in order to remove other cell types.” References: There are still some journal names which are given as full name and not as abbreviation.

Reviewer 3 Report

p.p1 {margin: 0.0px 0.0px 0.0px 0.0px; font: 12.0px 'Helvetica Neue'} p.p2 {margin: 0.0px 0.0px 0.0px 0.0px; font: 12.0px 'Helvetica Neue'; min-height: 14.0px}

The title of the manuscript is” Space Radiation induces long term impact on the Cardiovascular System by the activation of FYN through Reactive Oxygen Species.” This most important claim is not considered scientifically valid  for two reasons. First, it lacks appropriate time-matched control necessary to deny the effect of time rather than radiation on the altered gene expression. Second, there is no supporting evidence from the authors’ data that FYN activation affects the cardiovascular system through ROS.

About the first point, authors’ data itself shows alteration of the gene expression between days 1 and 3 in the 0Gy control in Figure 1E, which is supported by the difference of PC1 values in the controls in Figure 1B. The authors state “Unfortunately, the poor setup of the original experiment is out of our control.” However, it is authors’ responsibility to choose the right dataset and to design the appropriate research framework. They further state “The point of this study is to show how GeneLab datasets can be used to generate novel and interesting hypothesis to guide future space biology experiments.” However, it is difficult to support the point using “the poor setup of the original experiment.” Furthermore, they removed the data of days 7 and 28 as “outliers” in Figure 4 (lines 309-311). Then the data from these days should be removed from all the other analysis, and it would no longer be “long term impact.”

About the second point, the authors state “It has been shown from past research that potentially the majority of the ROS production for cardiac tissue is produced by the mitochondria.” However, the evidence in the literature will not guarantee or certify that the same thing happened to the authors’ case, much less about the "impact on the Cardiovascular System by the activation of FYN through Reactive Oxygen Species."